# Physically asymmetric division of the *C. elegans* zygote ensures invariably successful embryogenesis

Radek Jankele[1], Rob Jelier[2], Pierre Gönczy[1]*

[1]Swiss Institute for Experimental Cancer Research (ISREC), School of Life Sciences, Swiss Federal Institute of Technology Lausanne (EPFL), Lausanne, Switzerland; [2]Centre of Microbial and Plant Genetics, Katholieke Universiteit Leuven, Leuven, Belgium

**Abstract** Asymmetric divisions that yield daughter cells of different sizes are frequent during early embryogenesis, but the importance of such a physical difference for successful development remains poorly understood. Here, we investigated this question using the first division of *Caenorhabditis elegans* embryos, which yields a large AB cell and a small $P_1$ cell. We equalized AB and $P_1$ sizes using acute genetic inactivation or optogenetic manipulation of the spindle positioning protein LIN-5. We uncovered that only some embryos tolerated equalization, and that there was a size asymmetry threshold for viability. Cell lineage analysis of equalized embryos revealed an array of defects, including faster cell cycle progression in $P_1$ descendants, as well as defects in cell positioning, division orientation, and cell fate. Moreover, equalized embryos were more susceptible to external compression. Overall, we conclude that unequal first cleavage is essential for invariably successful embryonic development of *C. elegans*.

*For correspondence:
pierre.gonczy@epfl.ch

**Competing interests:** The authors declare that no competing interests exist.

## Introduction

Asymmetric divisions generate cell fate diversity during development and differentiation. Daughter cells of such divisions can differ not only in their asymmetric inheritance of fate determinants but also in their relative size. Such physically unequal asymmetric divisions are widespread during development. For instance, in *Caenorhabditis elegans* larvae, QR.a neuroblasts divide unequally owing to the asymmetric distribution of the non-muscle myosin II NMY-2, resulting in a larger daughter with a neuronal fate and a smaller daughter that undergoes apoptosis (*Ou et al., 2010*). Equalizing QR.a division through NMY-2 manipulation results in two cells adopting the neuronal fate (*Ou et al., 2010*). Likewise, *Drosophila* stem-like larval neuroblasts divide unequally to regenerate a larger neuroblast and a smaller ganglion mother cell that differentiates towards a neuronal fate. Experimentally induced size equalization of the neuroblast division prevents such differentiation despite proper asymmetric inheritance of the neuronal fate determinant Prospero (*Cabernard and Doe, 2009*; *Kitajima et al., 2010*). These examples illustrate how size differences can have a drastic consequence on the fate of resulting daughter cells. Physically unequal divisions are particularly prevalent during early embryogenesis in many systems, but the specific importance of size differences at this early stage for successful completion of development has been scarcely addressed.

Early embryogenesis in the nematode *C. elegans* entails several asymmetric divisions, including ones that yield daughter cells of different sizes (reviewed in *Rose and Gönczy, 2014*). The first of these is the unequal cleavage of the one-cell stage embryo (hereafter zygote) into the larger anterior cell AB and the smaller posterior cell $P_1$, corresponding respectively to ~60% and 40% of total embryo size. The variability in this size difference is minimal in the wild type (*Kemphues et al., 1988*), suggestive of functional importance. Further support for such importance comes from the

fact that unequal first cleavage occurs throughout the entire *Rhabditida* order, which is estimated to encompass an evolutionary distance as large as that between echinoderms and mammals (*Kiontke, 2005*; *Kiontke et al., 2011*), although the extent of size asymmetry varies between nematode species within this order (*Brauchle et al., 2009*; *Valfort et al., 2018*).

The unequal cleavage of the *C. elegans* zygote results from a rapid sequence of events that begins after fertilization (reviewed in *Rose and Gönczy, 2014*). First, symmetry of the zygote is broken by the centrosomes derived from the sperm (*Goldstein and Hird, 1996*; *Sadler and Shakes, 2000*). Thereafter, the contractile cortical acto-myosin network moves towards the future embryo anterior (*Munro et al., 2004*), accompanied by the establishment of mutually exclusive cortical domains of PAR polarity domains, with aPKC-3/PAR-3/PAR-6 at the anterior and PAR-1/PAR-2/LGL-1 at the posterior (reviewed in *Kemphues and Strome, 1997*; *Rose and Gönczy, 2014*). The PAR polarity network then ensures the asymmetric distribution of proteins and mRNAs through the action of polarity mediators, including the RNA-binding proteins MEX-5/6, which are present in an anterior–posterior (A-P) gradient prior to first cleavage (*Schubert et al., 2000*).

A-P polarity cues also direct asymmetric positioning of the mitotic spindle, which is located slightly off-center towards the posterior side by the end of anaphase, thus dictating unequal first cleavage (*Grill et al., 2001*). Asymmetric spindle positioning relies on an evolutionarily conserved ternary complex that anchors the molecular motor dynein at the cell cortex (reviewed in *Kotak and Gönczy, 2013*). In *C. elegans*, this ternary complex comprises the membrane-associated Gα proteins GOA-1 and GPA-16, the TPR and GoLoCo domain-containing proteins GPR-1/2, as well as the coiled-coil protein LIN-5, which interacts with a dynein motor complex (*Colombo et al., 2003*; *Gotta et al., 2003*; *Gotta and Ahringer, 2001*; *Srinivasan et al., 2003*). Dynein thus anchored at the cell cortex is thought to exert a pulling force on the plus-end of depolymerizing astral microtubules and, thereby, on the connected spindle poles. In response to A-P polarity cues, more GPR-1/2 and possibly LIN-5 are present on the posterior cortex, resulting in a larger net force pulling on the posterior spindle pole, leading to the unequal cleavage of the zygote into the larger AB cell and the smaller $P_1$ cell (*Colombo et al., 2003*; *Gotta et al., 2003*; *Grill et al., 2003*; *Grill et al., 2001*; *Park and Rose, 2008*; *Tsou et al., 2003*).

During subsequent stages of *C. elegans* embryogenesis, most cells derived from AB divide symmetrically and with nearly synchronous timing, whereas $P_1$ descendants undergo three additional asymmetric divisions, generating sister lineages with asynchronous division timing (*Sulston et al., 1983*). Overall, this leads to the generation of seven founder cells that will each yield specific tissues; for instance, E will give rise to the intestine and $P_4$ to the germline (*Sulston et al., 1983*). The fate of each founder cell is specified by asymmetrically distributed maternally provided components, together with regulated protein translation and degradation mechanisms operating in the embryo (reviewed in *Rose and Gönczy, 2014*; *Zacharias and Murray, 2016*). Global activation of zygotic transcription occurs at the ~26-cell stage (*Powell-Coffman et al., 1996*; *Schauer and Wood, 1990*), which is also when the onset of gastrulation is apparent by ingression of the E descendants Ea/Ep from the embryo periphery (reviewed in *Goldstein and Nance, 2020*). Whereas Ea/Ep descendants adopt an intestinal fate in a cell-autonomous manner, for other cells proper fate acquisition and behavior, including division orientation, relies on interactions among neighbors (reviewed in *Goldstein and Nance, 2020*; *Mango, 2009*; *Rose and Gönczy, 2014*). Overall, as a result of stereotyped division timing, cell positioning, division orientation, and fate allocation, the hermaphrodite *C. elegans* embryo invariably hatches with 558 cells (*Sulston et al., 1983*).

Despite the wealth of knowledge brought by decades of investigating developmental mechanisms in *C. elegans*, the specific importance of the size difference component of asymmetric division in the early embryo is not clear, including for the unequal division of the zygote. Investigating this question requires to specifically alter the size of the daughter cells resulting from first cleavage without altering A-P polarity or compromising subsequent cell divisions. This has been challenging: although *par* mutant embryos undergo equal first cleavage, A-P polarity is abolished entirely in these cases (*Kemphues et al., 1988*). Moreover, whereas depletion of *goa-1/gpa-16*, *gpr-1/2*, or *lin-5* by RNAi results in equal first division without A-P polarity impairment, such embryos exhibit severe cell division defects in subsequent cycles (*Colombo et al., 2003*; *Fisk Green et al., 2004*; *Gotta et al., 2003*; *Gotta and Ahringer, 2001*; *Lorson et al., 2000*; *Srinivasan et al., 2003*). Here, we specifically equalized AB and $P_1$ without altering A-P polarity or disrupting later cell divisions using acute

genetic inactivation or optogenetic manipulation of LIN-5, allowing us to address specifically the importance of the physical asymmetry of the *C. elegans* zygote division.

## Results

### Altering cell size asymmetry through manipulation of *lin-5* function

To address whether the size difference between AB and $P_1$ that derives from the unequal division of the *C. elegans* zygote is important for subsequent development, we took advantage of the temperature-sensitive *lin-5(ev571)* mutant allele (*Lorson et al., 2000*). We discovered that shifting *lin-5 (ev571)* zygotes from 17°C to 27°C for ~5 min during mitosis results in the spindle remaining in the cell center, yielding AB and $P_1$ cells of more similar sizes than normal (*Figure 1A–C*, *Figure 1—figure supplement 1*, *Videos 1–3*). Such embryos are referred to as 'upshifted *lin-5(ev571)*' hereafter. There was a slight variability in the size of AB relative to that of the entire zygote in these embryos (51.2%±2.8 standard deviation [SD], N = 257). Moreover, we found that inactivation of function was readily reversible since returning upshifted *lin-5(ev571)* embryos to 17°C after the ~5 min upshift rapidly restored LIN-5 function, allowing subsequent cell divisions to proceed (see below). Furthermore, to control for potential effects of the *lin-5(ev571)* mutation unrelated to AB and $P_1$ equalization, early 2-cell stage *lin-5(ev571)* embryos were shifted from 17°C to 27°C for ~5 min; such embryos are referred to as 'controls' throughout the article.

Severe or complete depletion of *lin-5* function does not impair A-P polarity (*Srinivasan et al., 2003*), and we found the same to be true upon acute LIN-5 inactivation in upshifted *lin-5(ev571)* embryos, as evidenced by the normal distribution of the posterior polarity protein PAR-2 (*Figure 1—figure supplement 2A, B*). Moreover, the polarity mediator MEX-5, which is enriched in the anterior cytoplasm of the wild-type zygote in response to A-P polarity cues (*Cuenca, 2003*; *Schubert et al., 2000*), was localized similarly in control and upshifted *lin-5(ev571)* embryos, resulting in MEX-5 enrichment in AB following first division (*Figure 1—figure supplement 2C*). In addition, as anticipated from the more centrally located first cleavage furrow, slightly more MEX-5 was present in $P_1$ in upshifted *lin-5(ev571)* embryos than in the control (*Figure 1—figure supplement 2D*). Taken together, these observations indicate that upshifted *lin-5(ev571)* zygotes exhibit normal A-P polarity, providing us with an experimental means to address specifically the impact of cell size asymmetry.

In search for a complementary method to equalize first division, we reasoned that optogenetic recruitment of LIN-5 to the anterior cortex could counteract the larger posterior forces normally acting on astral microtubules, thus maintaining the spindle in the center (*Fielmich et al., 2018*). We used an extant worm strain expressing both LOV::PH::GFP bound to the plasma membrane and endogenously tagged LIN-5::ePDZ::mCherry, which can thus be recruited to the desired portion of the plasma membrane by transient local induction of LOV<-> ePDZ interaction using 488 nm laser light (*Fielmich et al., 2018*; *Figure 1D, E*). As anticipated, we found that acute recruitment of LIN-5::ePDZ::mCherry to a small region of the anterior cortex can maintain the spindle in the center of the zygote, producing similarly sized AB and $P_1$ cells (*Figure 1F*, *Video 4*).

Overall, we conclude that two means of balancing anterior and posterior pulling forces during mitosis can serve to equalize the first division.

### First cleavage equalization can be tolerated up to a size asymmetry threshold

To assess the potential importance of the size difference between AB and $P_1$ for subsequent development, we utilized time-lapse microscopy to analyze upshifted *lin-5(ev571)* embryos and optogenetically manipulated embryos following return to their respective permissive conditions, scoring embryonic lethality at the motile 3-fold stage or at hatching (Materials and methods). We found that those upshifted *lin-5(ev571)* embryos that died invariably failed to elongate into a worm shape towards the end of embryogenesis, reflecting a failure in ventral closure of the epidermis that was accompanied by extrusion of endodermal and mesodermal tissues as muscle contraction began (*Figure 1C*, *Figure 1—figure supplement 3*, *Video 5*, *Video 6*; n = 92). The terminal embryonic phenotype of optogenetically manipulated embryos that died was likewise characterized by a failure in ventral epidermal closure, with the exception of one embryo that died as an unhatched larva (n = 26). Moreover, we found several post-embryonic phenotypes among upshifted *lin-5(ev571)*

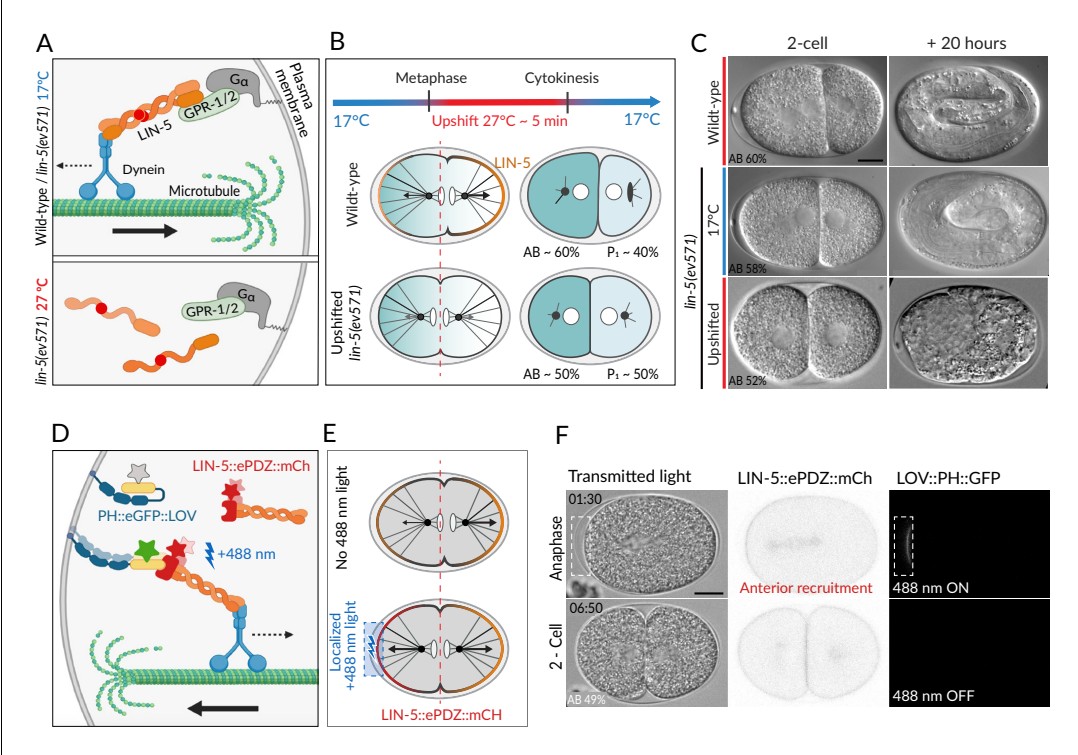

**Figure 1.** Two methods for equalizing first cell division in *C. elegans* embryos. (**A**) The ternary complex comprising Gα, GPR-1/2, and LIN-5 dimers tethers a dynein complex (only the motor protein is represented here) to the cell cortex, thereby generating pulling forces on astral microtubules (top). The *lin-5(ev571)* temperature-sensitive mutant encodes a protein with a 3-amino acid insertion (red disc) in the coiled-coil domain; this mutant protein cannot dimerize or generate pulling forces at the restrictive temperature (bottom). (**B**) Schematic of transient temperature upshift during the first division of the *C. elegans* zygote to equalize AB and P₁ sizes using *lin-5(ev571)*. Such transient upshift has no impact on the asymmetry of the first unequal division in the wild type, yielding AB and P₁ cells corresponding to ~60% and ~40% of embryo size, respectively (middle), due to asymmetric net cortical pulling forces acting on the spindle poles (arrows), stemming from the posterior enrichment of the ternary complex (orange). Cortical pulling forces are impaired upon transient upshift of *lin-5(ev571)* embryos, resulting in equalized first division (bottom). The red dashed line indicates the embryo center. Embryos are oriented with anterior to the left here and all figure panels. (**C**) Differential interference contrast (DIC) microscopy images from time-lapse recordings at the 2-cell stage (left) and 20 hr thereafter (right). Upshifted wild-type embryo (top) or *lin-5(ev571)* embryo kept at 17°C (middle) divide unequally and develop into 3-fold larvae that later hatch, while upshifting *lin-5(ev571)* embryos results in equalized division (bottom), which can lead to embryonic lethality. In this and other figure panels, scale bar is 10 μm and relative AB cell size at the 2-cell stage is indicated. See also *Videos 1–3*. (**D**) Schematic of optogenetic-mediated LIN-5::ePDZ::mCherry recruitment to the anterior cortex by localized activation of PH::eGFP::LOV interaction with a 488 nm laser. A helix in the LOV domain unfolds upon illumination, allowing binding of ePDZ. Star represents fluorescent protein fusion (GFP: gray to start with, green upon illumination; mCherry: red). (**E**) Embryos expressing LIN-5::ePDZ::mCherry and PH::eGFP:: LOV divide unequally without 488 nm exposure due to asymmetric localization of LIN-5 (top). Optogenetic-mediated recruitment of LIN-5::ePDZ:: mCherry to the anterior cortex during mitosis (blue rectangle) results in balanced net pulling forces on the two spindle poles and equal first division (bottom). (**F**) Images from time-lapse recording of optogenetic-mediated first division equalization. LIN-5::ePDZ::mCherry was recruited to the anterior cortex from anaphase onset until cytokinesis completion by scanning the 488 nm laser in the indicated rectangular region. Note that endogenous LIN-5 is tagged with ePDZ::mCherry, hence resulting in the fusion protein being present also on the spindle, the centrosomes, and the posterior cortex during anaphase. Note that that this strain also expresses GFP::TBB-2, which is not visible in the small illuminated portion of the cortex. Time indicated in min:s. See also *Video 4*.

The online version of this article includes the following source data, source code and figure supplement(s) for figure 1:

**Figure supplement 1.** Cell size measurement method and precision.
**Figure supplement 1—source code 1.** Comparison of 3D and 2D cell size measurements.
**Figure supplement 1—source data 1.** Matched 3D and 2D cell size measurements at the 2-cell stage.
**Figure supplement 2.** Upshifted *lin-5(ev571)* embryos are polarized.
**Figure supplement 2—source code 1.** Analysis of PAR-2::GFP and mCherry::MEX-5 intensities.
**Figure supplement 2—source data 1.** PAR-2::GFP and mCherry::MEX-5 intensities.
**Figure supplement 3.** Ventral rupture in equalized embryos.
**Figure supplement 3—source data 1.** Postembryonic phenotypes in individual animals.

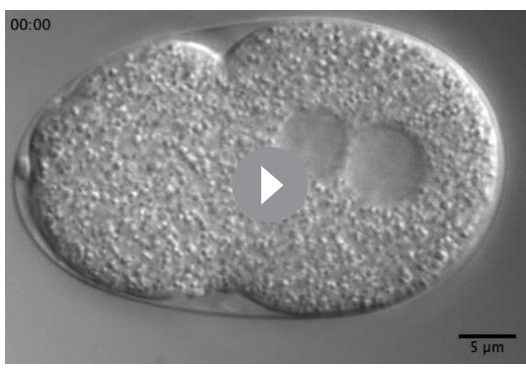

**Video 1.** Unequal division of the wild-type *C. elegans* zygote. The resulting 2-cell stage of this particular embryo had an AB size of 62% of the total embryo in the mid-plane. Note vertical rocking of spindle as it moves towards the posterior pole during anaphase. Time is indicated in min:s in this and all other videos.
https://elifesciences.org/articles/61714#video1

animals that hatched, including death during larval stages and reduced adult fertility (*Figure 1—figure supplement 3C*), demonstrating that equalizing the first division can result in lethality beyond embryogenesis.

We took advantage of the slight variability in relative AB size following equalization to ask whether this trait correlated with the incidence of embryonic lethality. As shown in *Figure 2A, B*, we observed that whereas control embryos exhibited nearly identical lethality compared to *lin-5(ev571)* embryos maintained at 17°C (~7%), lethality of upshifted *lin-5(ev571)* embryos gradually increased as AB and $P_1$ became closer in size, from ~24% when relative AB size was (56–54]% to ~75% when it was (50–48]% (refer to *Supplementary file 6* for statistical analyses throughout the article). We found a similar gradual increase of embryonic lethality when AB and $P_1$ became closer in size in optogenetically manipulated embryos (*Figure 2C, D*), which however exhibited ~23% lethality even when not exposed to 488 nm laser light. In addition, we found in both experimental settings that embryos with a relative AB size smaller than 48% invariably died (*Figure 2B, D*; 'inverted embryos' hereafter).

Taken together, these findings establish that first cleavage equalization can be tolerated to some extent since a subset of embryos with similar AB and $P_1$ sizes survive. However, other such embryos die, suggesting that the unequal first cleavage is required for invariably successful embryogenesis. Furthermore, our findings reveal a size asymmetry threshold at the 2-cell stage below which embryogenesis always fails.

## Decreased asynchrony between AB and $P_1$ upon first division equalization does not cause lethality of equalized embryos

We investigated further the consequences of rendering AB and $P_1$ similar in size. We utilized the *lin-5(ev571)* upshift setting for all experiments described hereafter because the optogenetic strain exhibited higher background lethality and was not amenable to investigation with fluorescently labeled markers since both GFP and mCherry channels are already occupied.

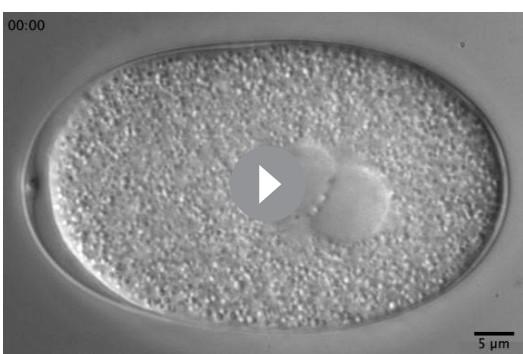

**Video 2.** Unequal division of the *lin-5(ev571)* mutant zygote at the permissive temperature of 17°C (relative AB size 58%). Note severely dampened anaphase spindle rocking indicating reduced cortical pulling forces in this video and in *Video 3*.
https://elifesciences.org/articles/61714#video2

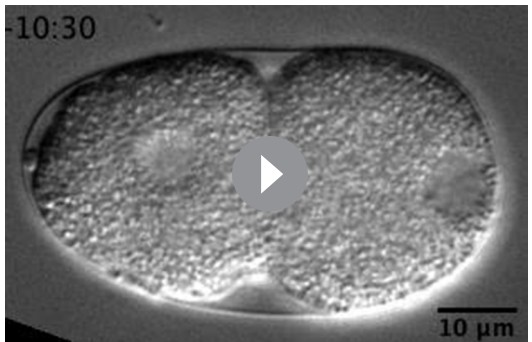

**Video 3.** Transient upshift of *lin-5(ev571)* embryo from 17°C to 27°C from metaphase until the completion of cytokinesis results in equalized or even inverted division, as in this particular embryo, in which AB is smaller than $P_1$ (AB size 47%).
https://elifesciences.org/articles/61714#video3

We first report our analysis of cell division timing. In the wild type, AB divides before $P_1$, in part owing to a size difference between the two cells (*Brauchle et al., 2003*). We addressed whether such a reduction in asynchrony was recapitulated in upshifted *lin-5(ev571)* embryos, using H2B::mCherry to precisely time anaphase onset. As shown in *Figure 2E*, we indeed found a strong correlation between relative AB size and extent of division asynchrony, confirming that the size difference between AB and $P_1$ contributes to their division asynchrony. Moreover, inverted embryos retained residual asynchrony (*Figure 2E*), verifying that cell size is only one of the factors dictating such asynchrony (*Budirahardja and Gönczy, 2008*; *Rivers et al., 2008*).

Could the reduction in division asynchrony between AB and $P_1$ cause the lethality of upshifted *lin-5(ev571)* embryos? We reasoned that if this were the case, then survival should be higher in upshifted *lin-5(ev571)* embryos in which asynchrony between the two cells was experimentally restored to the wild-type timing. Therefore, we set out to specifically slow down the cell cycle of $P_1$ in upshifted *lin-5(ev571)* embryos by directing a 405 nm laser onto its nucleus, adapting a method used previously for endodermal cells in the worm (*Lee et al., 2006*). We determined a non-lethal irradiation regime that retarded $P_1$ division timing in the wild type by 132 ± 94 s on average (n = 7), but did not cause lethality (Materials and methods). We found that a similar treatment in upshifted *lin-5(ev571)* embryos restored the extent of asynchrony between AB and $P_1$ to that of control embryos (*Figure 2F*). Despite this, however, viability was not rescued (*Figure 2F*), leading us to conclude that reduction in division asynchrony between AB and $P_1$ per se is not responsible for the lethality of upshifted *lin-5(ev571)* embryos.

Interestingly, in addition, we found a severe phenotype arising at the 4-cell stage in a minor fraction of upshifted *lin-5(ev571)* embryos, which we reasoned might be related to reduced division asynchrony between AB and $P_1$. In the wild type, division asynchrony results in the AB spindle elongating before the $P_1$ spindle, and thereby to an oblique positioning of the AB daughter cells due to the geometrical constraints imposed by the eggshell (*Figure 2—figure supplement 1A*). This leads to a characteristic rhomboid arrangement at the 4-cell stage, with the $P_1$ daughter $P_2$ contacting the AB daughter ABp (*Figure 2—figure supplement 1A*). $P_2$ expresses the Delta-like ligand APX-1, which interacts with the Notch receptor GLP-1 present on ABp, thereby instructing ABp descendants to adopt neural and epidermal fates (*Mango et al., 1994b*; *Mello et al., 1994*; *Shelton and Bowerman, 1996*; *Figure 2—figure supplement 1B*). We found that ~4% of upshifted *lin-5(ev571)* embryos adopted an abnormal T-like arrangement at the 4-cell stage (T-arrangement thereafter) (*Figure 2—figure supplement 1C*; *Video 7*; n = 257). In such embryos, $P_2$ cannot contact ABp and, by inference, Delta/Notch signaling cannot take place between the two cells (*Figure 2—figure supplement 1D*). As anticipated from such lack of signaling, all embryos with a T-arrangement at the 4-cell stage later died (n = 11).

Overall, we conclude that reduced division asynchrony between AB and $P_1$ per se is not responsible for the lethality of most upshifted *lin-5(ev571)* embryos. Moreover, we find that a

**Video 4.** Optogenetic recruitment of transgenic LIN-5 to the anterior cortex during mitosis counteracts posterior forces and results in equalized division (relative AB size 49%). Montage showing DIC recording (left), LIN-5::EPDZ::mCherry (middle), and PH::EGFP::LOV (right). LIN-5::EPDZ::mCherry was recruited to membrane-bound PH::EGFP::LOV by exposing a small rectangular region at the anterior cortex with a 488 nm laser during mitosis.

https://elifesciences.org/articles/61714#video4

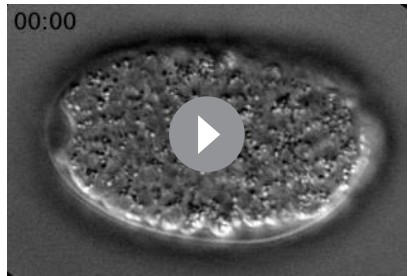

**Video 5.** Ventral closure of epidermis at the end gastrulation in control *lin-5(ev571)* embryo at the permissive temperature of 17°C. Lateral cells migrate over ventral neuroblasts to enclose the embryo in a continuous epidermal layer called the hypodermis. Subsequently, differentiating muscles start to contract and the embryo gradually elongates into a tubular body shape.

https://elifesciences.org/articles/61714#video5

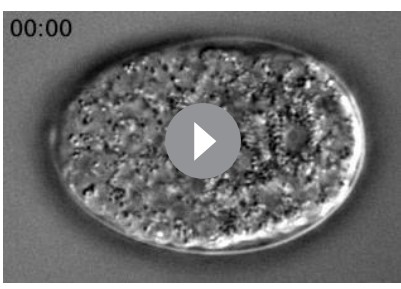

**Video 6.** Upshifted equalized *lin-5*(*ev571*) embryo failing to undergo ventral epidermis closure, resulting in the extrusion of internal tissues from the body cavity as muscles start contracting. Embryo shown here had a relative AB size of 53%.
https://elifesciences.org/articles/61714#video6

minor fraction of upshifted *lin-5*(*ev571*) embryos lack contact between $P_2$ and ABp and, thereby, a critical Delta/Notch induction event, illustrating how the size difference between AB and $P_1$ can contribute to successful fate acquisition.

## Comprehensive 4D cell lineaging to monitor development of equalized embryos

To reach a more comprehensive understanding of potential deviations from normal development following first division equalization, we conducted systematic 4D lineage tracing using nuclear H2B::mCherry as a proxy for cell position (*Bao et al., 2006*; *Jelier et al., 2016*; *Krüger et al., 2015*). We collected data up to the ~120 cell stage, assessing 13 features for each cell, including division timing, cell position, and division orientation, resulting in up to ~2000 data points per embryo, although not all stages or features could be scored in every instance (see *Supplementary file 1* for information regarding each lineaged embryo and *Supplementary file 2* for individual cell data). The entire lineaging data set is available at https://github.com/UPGON/worm-rules-eLife; (*Jankele, 2021*; copy archived at swh:1:rev:069c5e3147b7721885b5824282f342ca-c8a4de5b). The lineaging data set comprised wild-type embryos (wild-type, n = 10), *lin-5*(*ev571*) embryos shifted early in the 2- cell stage (control, n = 18), as well as upshifted *lin-5*(*ev571*) embryos that were equalized (empirically defined as having a relative AB size of 48–53%), and that either lived (equalized alive, n = 21) or died (equalized dead, n = 28). There was no significant difference in relative AB size between these two equalized groups (50.6% in both; p=0.99), so that their distinct fate later in embryogenesis cannot be attributed to a size difference at the 2-cell stage. In addition, we lineaged *lin-5*(*ev571*) embryos with a <48% relative AB size, all of which died (inverted, n = 7). Given their severe and early departure from normality, we did not lineage the minor fraction of *lin-5*(*ev571*) embryos with a 4-cell stage T-arrangement.

We set out to mine this rich data set to uncover deviations from normal development following AB and $P_1$ equalization, comparing wild-type and control embryos with equalized *lin-5*(*ev571*) and inverted *lin-5*(*ev571*) embryos. In addition, we investigated whether differences could be unveiled between equalized alive and equalized dead *lin-5*(*ev571*) embryos to identify features that may cause lethality in the latter group.

## Lineaging reveals global alteration in cell cycle progression in the $P_1$ lineage and an altered sequence of cell division in equalized embryos

We first report our analysis of temporal aspects in the lineaging data. Plotting separately the proliferation of cells descending from AB and $P_1$ (hereafter referred to as AB and $P_1$ lineage, respectively), we observed no statistically significant difference in division timing in the AB lineage between wild-type or control and equalized or inverted *lin-5*(*ev571*) embryos (*Figure 3A* – overlapping confidence intervals, *Supplementary file 2*). In stark contrast, we found a clear difference in the $P_1$ lineage, where cell number increased more rapidly in equalized and in inverted embryos compared to wild-type or control embryos (*Figure 3B*). Furthermore, cell cycle duration in equalized *lin-5*(*ev571*) embryos was rarely correlated with relative AB size for AB descendants, but highly correlated for most $P_1$ descendants (*Figure 3—figure supplement 1*, *Supplementary file 3*).

Investigating cell cycle duration in individual blastomeres revealed that the germline precursor $P_4$ was most affected, with the cell cycle being ~33% shorter in equalized embryos compared to the control (*Figure 3C*, p<0.001 with Welch's two-sample t-test, which is used also hereafter unless stated otherwise). Interestingly, we also uncovered a deviation from the normal division pattern in the $P_4$ lineage suggestive of fate transformation. Whereas $P_4$ normally undergoes a single division (*Figure 3D*, top, *Figure 3E*), we found that it underwent at least one additional division in ~35% of equalized *lin-5*(*ev571*) embryos (n = 36), and in all four inverted *lin-5*(*ev571*) embryos scored for this

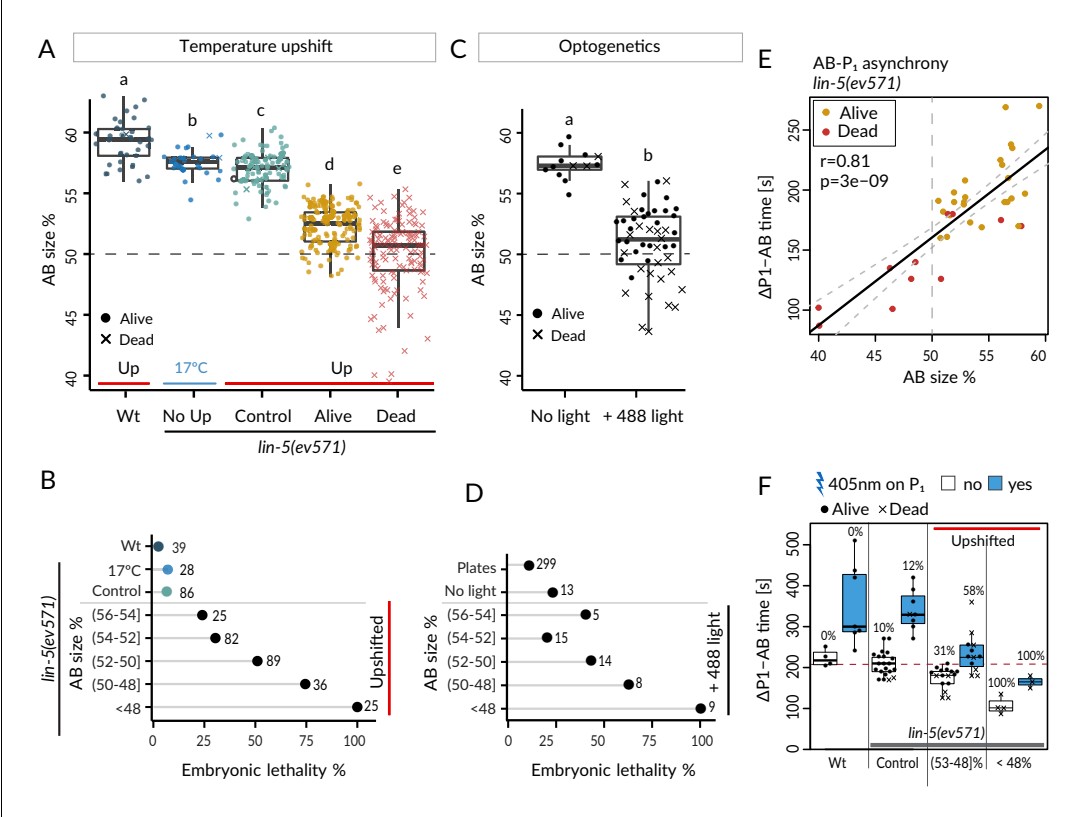

**Figure 2.** Equalized first cell division decreases embryonic viability. (**A**) Boxplots of relative AB size distribution determined in the mid-plane (here and thereafter) in embryos of the indicated conditions; upshifted (Up) wild-type (Wt), *lin-5(ev571)* kept at 17°C (No Up), *lin-5(ev571)* upshifted in the early 2-cell stage (Control), as well as *lin-5(ev571)* upshifted during the first mitotic division and binned into embryos that later lived (Alive) or died (Dead). Here and in other figure panels, two boxplots that do not share the same letter are significantly different from each other with p<0.05; Welch's t-test. See *Supplementary file 6* for exact p value and complete statistical analyses for this and all other figures. Dashed lines here and in **C** indicate equal AB and $P_1$ sizes. (**B**) Lethality of embryos of from **A**, with Alive and Dead categories binned as a function of relative AB size and an indication of the number of embryos in each bin. Brackets indicate a mathematical notation for the size interval, [with an inclusion of the value next to the square bracket], (but not of that next to the round bracket), so that individual bins do not overlap. (**C**) Relative AB size in embryos expressing LIN-5::ePDZ::mCherry and PH:: eGFP::LOV, either not subjected to 488 nm laser light (left, n = 13) or equalized through optogenetic recruitment of LIN-5 to the anterior cortex (right, n = 56). (**D**) Lethality of embryos from **C** as a function of relative AB size; binning as in **B**. 'Plates' refers to embryos expressing LIN-5::ePDZ::mCherry and PH::eGFP::LOV scored for hatching on plates, without filming. (**E**) Division asynchrony between AB and $P_1$ as a function of AB size in upshifted *lin-5 (ev571)* embryos; time was determined using anaphase onset monitored with mCherry::H2B. Shown is the subset of *lin-5(ev571)* control and upshifted embryos from **A** for which AB-$P_1$ asynchrony was measured. Pearson's correlation value (r) is shown here and in subsequent figure panels, together with significance of association between two variables determined via the asymptotic *t* approximation (cor.test function in R, p). (**F**) Division asynchrony between AB and $P_1$ in embryos of indicated conditions; upshifted *lin-5(ev571)* embryos were split into two bins as a function of AB sizes, as shown. Percentage above each boxplot indicates proportion of embryonic lethality; blue: illumination of $P_1$ with 405 nm laser to delay cell cycle progression and restore asynchrony between AB and $P_1$. Non-illuminated embryos: same data as in **E**. Dashed line indicates mean time difference in control embryos. See also *Figure 2—source data 1* and *Figure 2—source code 1* for underlying analysis and data for this figure.

The online version of this article includes the following source data, source code and figure supplement(s) for figure 2:

**Source code 1.** Cell size manipulation and lethality analysis in lin-5(ev571) and optogenetic embryos.
**Source data 1.** Cell size measurements and division timing in *lin-5(ev571)* and optogenetic embryos.
**Figure supplement 1.** Aberrant blastomere arrangement at the 4-cell stage.

trait (*Figure 3D*, bottom, *Figure 3E*). Overall, we conclude that following first division equalization the cell cycle is shortened most in the germline precursor $P_4$, the fate of which is also imparted less faithfully.

We found in addition that faster cycling in the $P_1$ lineage of equalized embryos eventually changed the relative order of cell division during development (*Figure 3F*). Notably, the division of the $P_1$ descendants Ea/Ep, D, Da, and $P_4$ occurred before that of the corresponding AB lineage cells

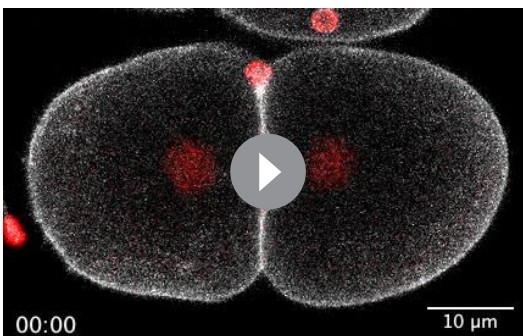

**Video 7.** Equalized size of AB and $P_1$ leads to an aberrant T-arrangement at the 4-cell stage in ~4% of equalized embryos (relative AB size is 50% in this case). https://elifesciences.org/articles/61714#video7

(*Figure 3F*, red arrows). Could such an altered sequence of events in the AB versus the $P_1$ lineage contribute to the lethality of equalized *lin-5 (ev571)* embryos? To address this question, we investigated whether there was a statistically significant difference in cell cycle duration or division timing comparing individual cells in equalized alive and equalized dead embryos, but found none (*Supplementary file 2*, Welch's t-tests with Benjamini–Hochberg correction for multiple tests). Taken together, our data demonstrate that equalizing the first division results in a global increase in cell cycle pace in the $P_1$ lineage, leading to an altered sequence of divisions, but that this alone does not explain why some equalized embryos die whereas others do not.

We were also interested in exploring the variability of cell cycle duration in the lineaging data set. We determined the coefficient of variation (CV) for each cell and found that the overall average CV of cell cycle duration was 4.4% ± 1.8 in the wild-type and 5.3% ± 1.8 in *lin-5(ev571)* control embryos (*Figure 3G*, p<0.001), indicative of increased variability already in the latter group. In addition, we found that whereas the CV in cell cycle duration in equalized alive *lin-5(ev571)* embryos was comparable to that of control embryos (5.2% ± 2.4, p=0.75), it was significantly larger than that in equalized dead and inverted *lin-5(ev571)* embryos, a difference due strictly to the $P_1$ lineage (*Figure 3G*, 6.6% ± 2.7% and 6.5% ± 4.2, respectively, p<0.0001). We conclude that equalized dead and inverted *lin-5(ev571)* embryos exhibit less stereotyped cell cycle durations.

## Lineaging reveals that equalized embryos frequently exhibit defects in cell positioning and division orientation

We next report our analysis of spatial aspects in the lineaging data. In particular, we monitored cell position and division orientation along the three embryonic axes, comparing their averages to a reference model of spatially and temporally aligned control *lin-5(ev571)* embryos (*Jelier et al., 2016*) (Materials and methods).

We found that equalized alive *lin-5(ev571)* embryos exhibited a mild increase in the average positional deviation per cell from the control starting at the 4-cell stage; this deviation remained essentially constant thereafter (*Figure 4A*, compare blue and yellow). By contrast, cell positions in equalized dead *lin-5(ev571)* embryos diverged increasingly from the control over time (*Figure 4A*, compare blue and red). For inverted embryos, cell positions deviated already early on and diverged further thereafter (*Figure 4A*, gray). Furthermore, we found that division orientation of individual cells exhibited a mild increase in angular deviation in equalized alive *lin-5(ev571)* embryos compared to the control, and that this deviation was more pronounced in equalized dead and inverted *lin-5 (ev571)* embryos (*Figure 4B*). Together, these findings establish that embryos that die following first division equalization diverge more from the norm in terms of cell positioning and division orientation than those that live.

## Alterations in EMS and MS division orientation contribute to lethality of equalized embryos

Investigating spatial features in the lineaging data for individual blastomeres revealed a striking skew in the division axis of EMS at the 6-cell stage in a subset of embryos. In the wild type, the EMS spindle is oriented along the A-P axis owing to Wnt/Src signaling emanating from $P_2$ (*Bei et al., 2002*; *Liro and Rose, 2016*). We found that EMS division orientation was skewed by >35° in ~19% equalized and ~43% inverted *lin-5(ev571)* embryos (*Figure 4F*; n = 48 and 7). In all eight embryos with such a skew that later died, this alteration led to striking cell mispositioning at the 8-cell stage, with E and MS assuming abnormal oblique positions (*Figure 4G*, top), a defect that propagated subsequently (*Figure 4G*, bottom). By contrast, in the four equalized embryos that lived despite an

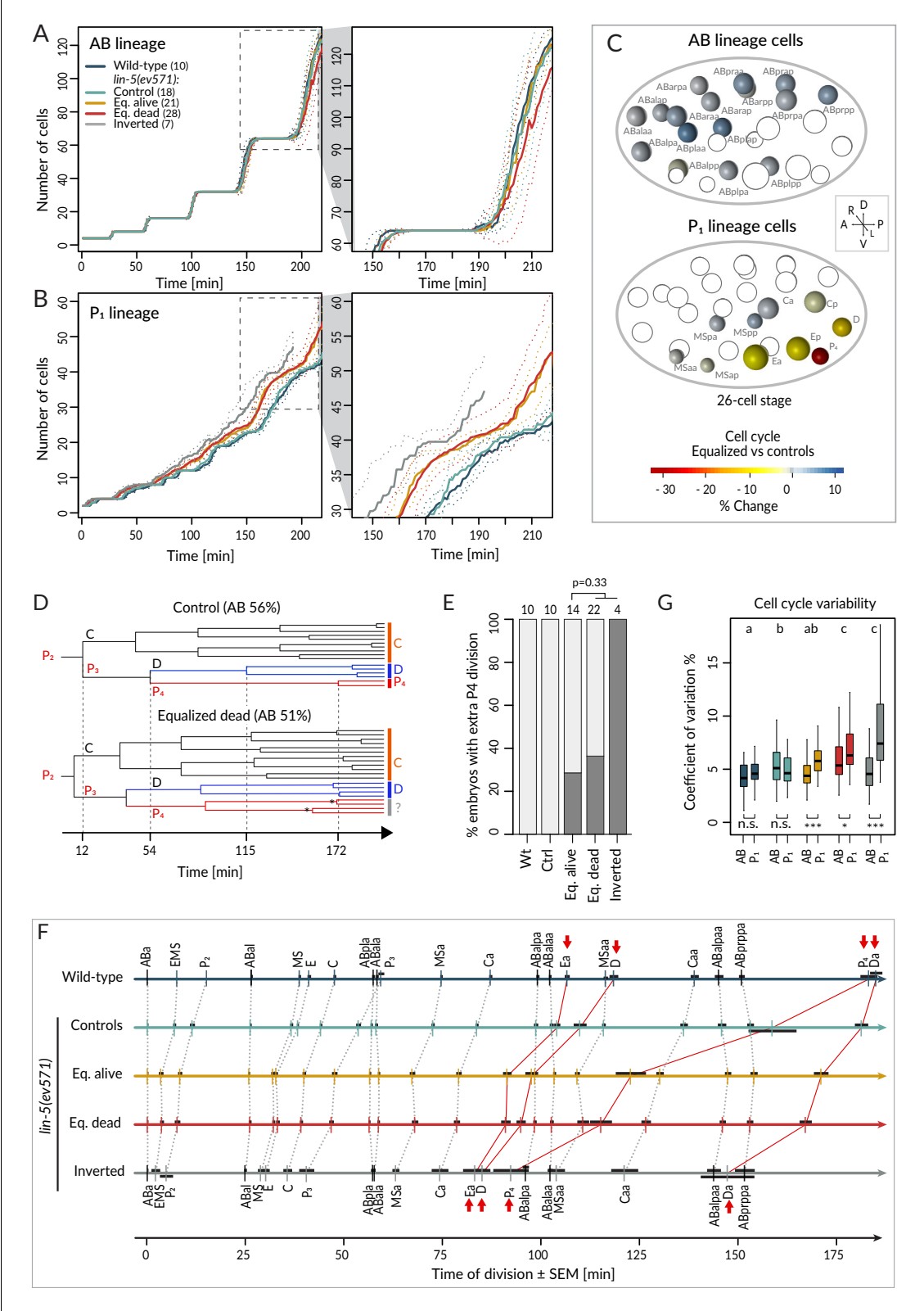

**Figure 3.** Faster cell cycle progression in P₁ descendants of equalized embryos results in altered division sequence. (**A, B**) Number of cells in the AB (**A**) and P₁ (**B**) lineages starting from the 4-cell stage in embryos of indicated conditions. Dashed rectangular region is enlarged on the right. In this and subsequent figure panels, time 0 corresponds to ABa cleavage. Dotted lines – most visible in the enlargements on the right – indicate standard deviations. (**C**) Graphical depiction of differences in cell cycle timing in individual cells of the AB (top) and P₁ (bottom) lineages at the 26 cell stage;

*Figure 3 continued on next page*

*Figure 3 continued*

colors indicate relative change of cell cycle timing in equalized *lin-5(ev571)* embryos compared to control *lin-5(ev571)* condition. In each representation, empty circles represent cells of the other lineage. (D) Partial lineage tree of a control *lin-5(ev571)* embryo (top) and an equalized *lin-5(ev571)* embryo (bottom). Normally, $P_4$ divides only once to produce the germline precursors Z2 and Z3, which remain quiescent until hatching (top). In some equalized and inverted embryos, $P_4$ divides more than once, suggesting a repetition of the $P_3$ fate or a fate transformation towards a D-like state (bottom, labeled by question mark). Also, a global acceleration of the C and D lineages is apparent in the equalized embryo compared to the control. Vertical grid lines and time indicated on the x axis correspond to division of $P_2$, $P_3$, D, and $P_4$ in the control embryo. (E) Frequency of embryos with one or more extra divisions in the $P_4$ lineage in indicated conditions (dark gray), with number of analyzed embryos shown on top. Light gray corresponds to embryos with a normal phenotype here and in similar panels in other figures. Ectopic P4 lineage divisions are not significantly more frequent in dying embryos than in those that are alive (Fisher's exact test, p=0.33). (F) Temporal division sequence in embryos in indicated conditions. Note that the faster cell cycle of $P_1$ lineage cells in equalized and inverted embryos leads to an altered division sequence, highlighted by connecting red lines and arrows for Ea, D, $P_4$, and Da. Dark horizontal bars indicate standard error of the mean. Not all AB cells are shown due to space constrains, but the first and last AB cell in each division round are indicated. (G) Variability in cell cycle duration in AB and $P_1$ lineage expressed as variation coefficient to normalize for cell cycle duration in different cells. Color code as in A. Here and in *Figure 4C, D*, letters above groups indicate statistical comparison of overall variability among five groups using ANOVA and Tukey's honest significant difference post-hoc test for all pairwise combinations, whereas comparisons between AB and $P_1$ lineages within groups are indicated below boxplots using Welch's two-sample t-test; asterisks indicate significance levels at 0.05 (*) or 0.001 (***).

The online version of this article includes the following figure supplement(s) for figure 3:

**Figure supplement 1.** Correlation of cell cycle with AB size in *lin-5(ev571)* embryos.

initially aberrant EMS division angle (*Figure 4F*), the skew was corrected in late anaphase, resulting in essentially normally positioned E and MS. These observations indicate that, unless corrected, a skew in the EMS division axis results in embryonic lethality.

To uncover further differences between equalized alive and equalized dead *lin-5(ev571)* embryos, excluding embryos with a T-arrangement at the 4-cell stage and those with an EMS skew at the 6-cell stage, we compared all spatial features between the two sets of embryos from the 4- to the 15-cell stage (*Figure 4—figure supplement 1A*, *Supplementary file 4*). This analysis uncovered 10 features that differed significantly between the two groups and that were highly correlated with each other (*Figure 4—figure supplement 1A*). Among these features, we found notably that the division orientation of MS deviated approximately three times more with respect to the A-P axis in equalized dead *lin-5(ev571)* embryos than in equalized alive ones (*Figure 4H*). This led to increased mispositioning of the MS daughter cells MSa and MSp (*Figure 4I*), which were frequently inverted along the L-R axis, a phenotype significantly associated with death (*Figure 4J, K*). However, such L-R inversion was also observed in some equalized alive *lin-5(ev571)* embryos (*Figure 4I*), indicating that it can be compatible with successful embryogenesis, presumably owing to subsequent compensatory movements. Taken together, these observations lead us to conclude that severe mispositioning of E and MS, as well as of MSa and MSp, can occur in equalized dead *lin-5(ev571)* embryos and contribute to lethality.

Like for the temporal data, we explored the overall variability of cell positioning and division orientation in the lineaging data set, comparing the SDs of all individual cells. As shown in *Figure 4C*, we found a slight increase in overall variability of cell positioning among equalized alive *lin-5*(ev571) embryos compared to the control, as well as a further significant increase among equalized dead and inverted *lin-5*(ev571) embryos (*Figure 4D*). Together, these results indicate that cell positioning and division orientation defects in equalized embryos are also variable rather than stereotyped, especially in embryos that die.

Serendipitously, we found that 6 of the 10 features identified above correlate with the extent of compression the embryo experienced from the coverslip during time-lapse imaging, suggesting that this external factor affects cell positioning, division angles, and cell migration differently in equalized alive and dead *lin-5(ev571)* embryos. Furthermore, equalized dead *lin-5(ev571)* embryos tended to be significantly more compressed than equalized alive ones (*Figure 4E*, 19.8 ± 2.1 μm versus 21.9 ± 2.2 μm, p<0.002), whereas a comparable degree of compression had no detrimental effect on control embryos (*Figure 4E*). These observations led us to investigate directly whether equalized embryos die less frequently in the absence of compression. Therefore, we imaged uncompressed embryos using 45 μm diameter beads or removing the coverslip after first division equalization, finding decreased lethality of equalized embryos compared to imaging under mild compressed

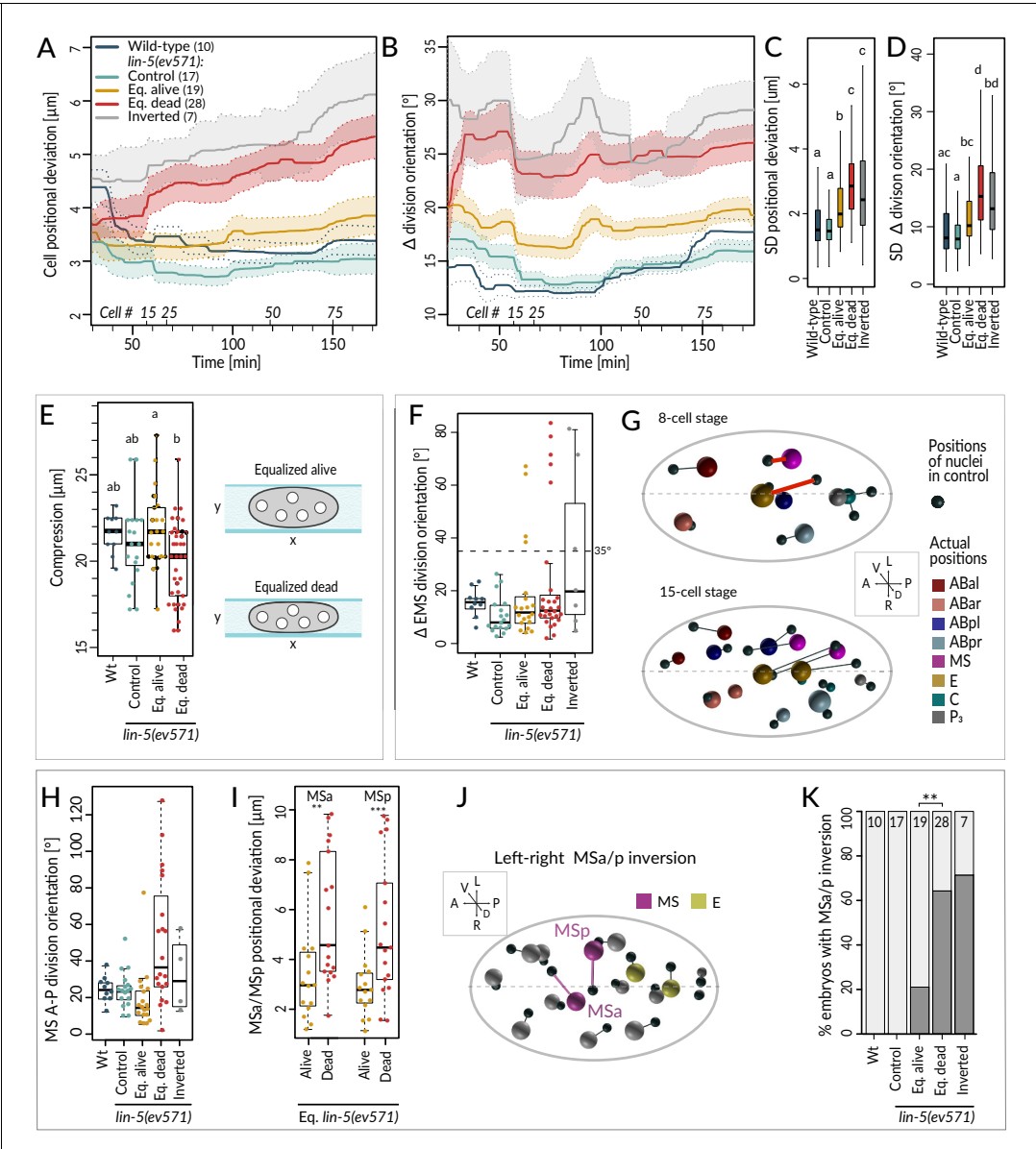

**Figure 4.** Cell division axis and cell position are frequently altered in equalized embryos. (A, B) Mean positional deviation (A) or angular deviation (B) ± SD per cell over time in embryos of indicated conditions compared to control *lin-5(ev571)* reference. The mean positional deviation per cell was calculated at 1 min intervals as the sum of Euclidean distances of individual cells divided by the number of cells at that stage. Only embryos lineaged past the 15-cell stage were included in this analysis. (C, D) Variability in cell position deviation (C) and angular deviation of division orientation (D) in embryos of indicated conditions expressed as SD for cells between 25 and 180 min of development (time 0 corresponds to ABa division). (E) Compression (sample height) in embryos of indicated conditions. Comparisons between groups were performed using Tukey's honest significant difference post-hoc test. (F) Angular deviation of EMS spindle at anaphase onset in embryos of indicated conditions compared to the control *lin-5 (ev571)* reference. Note that the four live equalized embryos exhibiting a >35° EMS skew at anaphase onset corrected spindle orientation to near normality during late anaphase. (G) Nuclear positions in the control *lin-5(ev571)* reference shown with black spheres, connected by lines with the position of the corresponding nuclei in an equalized *lin-5(ev571)* embryo exhibiting the EMS skew; sphere size is proportional to nuclear diameter. The skew in EMS division leads to mispositioning of E and MS at the 8-cell stage (highlighted by red line; top), resulting in widespread positioning defects at the 15-cell stage (bottom) and thereafter (not shown). Here and in J, dashed line indicates center of the L-R axis. The directions of the three embryonic axes are depicted here and in other panels. (H) Deviation of MS division orientation angle in embryos of indicated conditions compared to the control *lin-5(ev571)* reference. (I) Mispositioning of MSa and MSp in equalized *lin-5(ev571)* embryos measured as the distance to corresponding positions in the control *lin-5(ev571)* reference. (J) Positions of nuclei in equalized *lin-5(ev571)* embryo with L-R inversion of MSa and MSp (MSa and MSp: magenta; Ea and Ep: yellow), connected by lines to cell positions in the control *lin-5(ev571)* reference, indicated as small black spheres. (K) Quantification of MSa/MSp inversion phenotype in indicated conditions scored prior to MSa/MSp division; the phenotype is significantly associated with the outcome of development among equalized *lin-5(ev571)* embryos (Fisher's exact test, p=0.007).

*Figure 4 continued on next page*

*Figure 4 continued*

The online version of this article includes the following figure supplement(s) for figure 4:

**Figure supplement 1.** Systematic analysis of features in lineaged embryos up to 15-cell stage.

---

condition with 20–25 µm beads or in the continued presence of the coverslip (33% versus 55%, n = 45 and 174, p=0.03, Fisher's exact test). Together, these findings indicate that equalized embryos tolerate less well mechanical compression, which may perturb cell positioning, division orientation, and cell–cell contacts.

## Defective cell fate allocation in equalized embryos

We investigated whether defects in cell fate might occur following first division equalization, focusing our analysis on the endodermal and pharyngeal lineages.

In the wild-type endodermal lineage, the $P_1$-derived Ea/Ep cells exhibit a nearly twofold lengthening of the cell cycle compared to the MS lineage sister cells as they ingress into the blastocoel (*Figure 5A, B*, *Sulston et al., 1983*). Interestingly, we found that Ea/Ep divided earlier and ingressed less in equalized dead *lin-5(ev571)* embryos than in either control or equalized alive *lin-5(ev571)* embryos, proportionally to relative AB size (*Figure 5C, D*, *Figure 5—figure supplement 1A, B*, *Video 8*). We noted also that Ea/Ep tended to divide while still in contact with the eggshell in most equalized embryos (*Figure 5E*, green nuclei and dashed cell contour). Overall, we conclude that endodermal cells divide precociously and ingress only partially in equalized *lin-5(ev571)* embryos, particularly in the subset that later dies.

We set out to assess whether these alterations in Ea/Ep behavior are accompanied by improper endodermal fate specification. Normally, such specification is imparted by a redundant cascade of GATA-family transcription factors, including END-3/END-1, which peak in Ea/Ep and activate ELT-7/ELT-2, with ELT-2 remaining present throughout embryogenesis (reviewed in *Maduro, 2015*; *McGhee, 2013*). Suggestively, we found that END-3::GFP in equalized *lin-5(ev571)* embryos was expressed at lower levels than in controls (~62 ± 19%, p<0.001), correlating with relative AB size and Ea/Ep cell cycle duration (*Figure 5E, F*; n = 18 and 19, respectively, *Figure 5—figure supplement 1C*, *Video 8*). Furthermore, ELT-2::GFP, which is normally expressed in eight intestinal progenitors at the ~100-cell stage, was often expressed in only 2–6 cells in equalized *lin-5(ev571)* embryos (*Figure 5G, H*). Together, these findings indicate that endodermal fate is acquired less faithfully following first division equalization, although altered ELT-2::GFP distribution is not significantly associated with embryonic lethality (p=0.69, Fisher's exact test), in line with the notion that even embryos with a partial gut can survive (*Choi et al., 2017*).

We also investigated fate specification of the pharyngeal lineage. In the wild type, the anterior pharynx is induced in ABalp and ABara through Delta/Notch signaling stemming from MS/MSa/MSp (reviewed in *Mango, 2009*). Although all four ABa descendants express the GLP-1 Notch receptor, normally ABala and ABarp are not induced since they do not contact the MS/MSa/MSp cells that express the Delta ligand (*Figure 5—figure supplement 1D*). We analyzed embryos expressing the pharyngeal marker PHA-4::GFP, which is detectable at the ~100-cell stage in the lineages of MS, ABalp, and ABara, labeling 18 cells in total (*Horner et al., 1998*; *Mango et al., 1994a*; *Murray et al., 2008*). We found that equalized alive *lin-5(ev571)* embryos systematically harbored PHA-4::GFP in 18 nuclei (*Figure 5I, J*; n = 5). By contrast, abnormal PHA-4::GFP expression was observed in ~47% of equalized dead and inverted *lin-5(ev571)* embryos, usually with >18 positive nuclei (*Figure 5J*, *Figure 5—figure supplement 1E*; n = 15). Ectopic PHA-4::GFP was present in some cells derived from ABala in five such embryos and in ABarp in another one (*Figure 5—figure supplement 1E*), suggesting that MS/MSa/MSp formed illegitimate contacts in these cases, leading to aberrant pharynx induction in their descendants.

Overall, we conclude that equalized *lin-5(ev571)* embryos exhibit defective fate acquisition at the least in the endodermal and pharyngeal lineages.

## Viability of equalized embryos can be predicted at the 15-cell stage

We explored whether the above described defects occurred simultaneously in equalized embryos. However, we found no significant association between any pair of qualitative features described thus

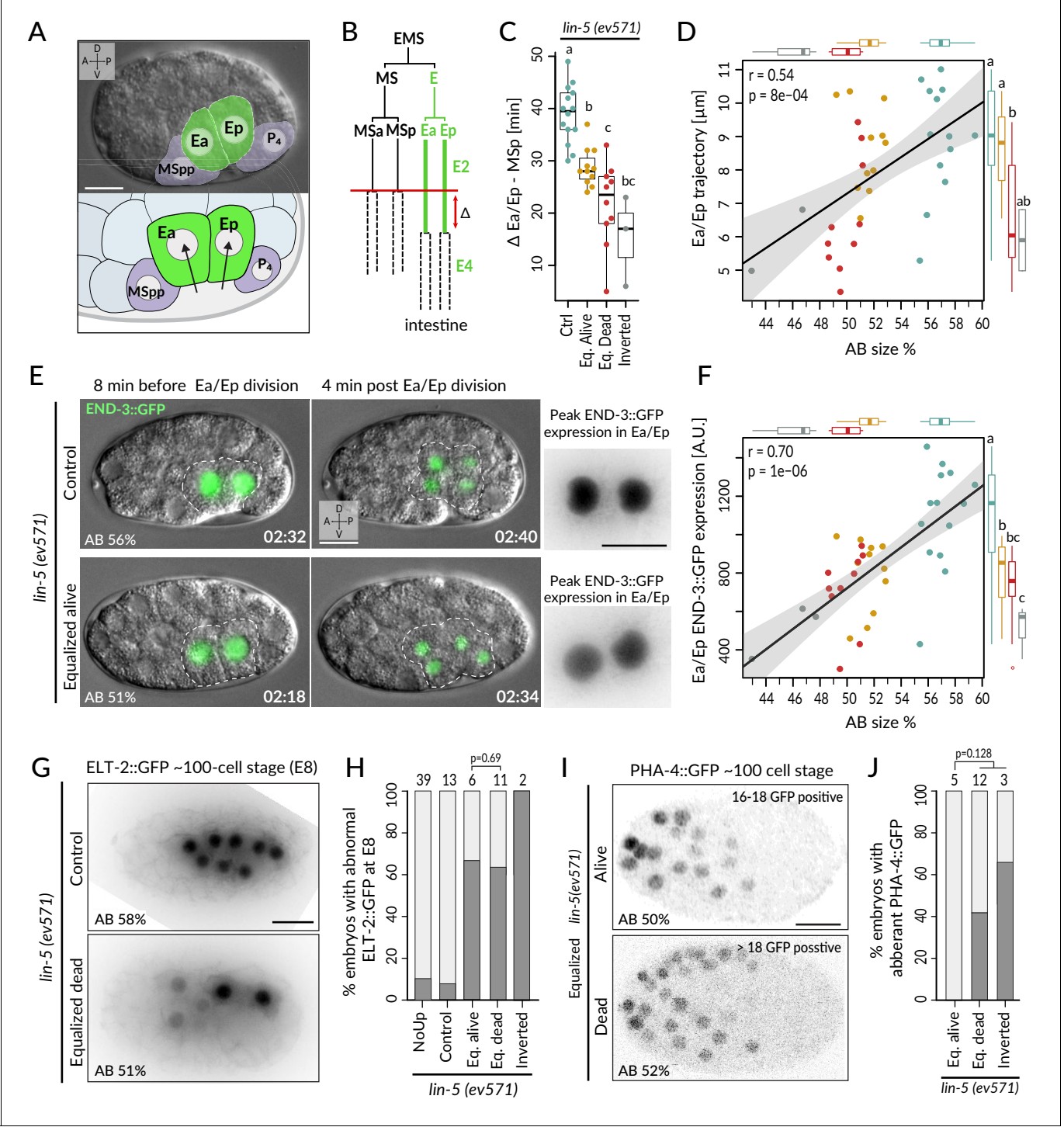

**Figure 5.** Incomplete fate acquisition in endodermal and pharyngeal lineages in equalized *lin-5(ev571)* embryos. (**A**) Gastrulation in *C. elegans* begins with ingression of Ea/Ep. Actual embryo viewed by DIC with partial overlay (top) and enlarged corresponding schematic (bottom). (**B**) Gastrulation is accompanied by a nearly twofold lengthening of the Ea/Ep cell cycle compared to that of their MSa/MSp cousins. Ea/Ep normally divide after internalization. Red arrow indicates usual delay between MSp and Ea/Ep divisions shown in C. (**C**) Time delay between MSp and Ea/Ep divisions in embryos of indicated conditions. (**D**) Average trajectory of Ea/Ep interphase nucleus as a function of initial AB size. Here and in F: same color code as in C; moreover, marginal boxplots show overall distribution of individual groups of corresponding color, with letters indicating whether the mean differs statistically between groups (Welch's t-test, p<0.05). (**E**) END-3::GFP expression in Ea/Ep (dashed area) in control and equalized *lin-5(ev571)* embryo; Ea/Ep cells in the latter divide close to the eggshell, before completing ingression. Higher magnification views on the right show the time point with peak END-3::GFP expression. Time is indicated in h:min from the time of first cleavage. See also *Video 8*. (**F**) Quantification of END-3::GFP peak

*Figure 5 continued on next page*

*Figure 5 continued*

expression during the Ea/Ep cell cycle as a function of initial AB size. Statistical comparisons are indicated using the letters code above the marginal boxplot (Welch's test, BH-corrected p-value<0.05). See also *Figure 5—source data 1* and *Figure 5—source code 1* for analysis for **C–F**. (**G**) Control (top) or equalized dead *lin-5(ev571)* (bottom) ~100-cell stage embryo expressing GFP::PH (not shown) and ELT-2::GFP in inverted black and white rendition. Note that the image of the control embryo has been rotated, explaining the white areas in the corners. (**H**) Quantification of embryos with abnormal ELT-2::GFP expression (defined as <8 GFP-positive cells at the ~100-cell stage) in indicated conditions. Lethality of equalized embryos is not significantly associated with ELT-2::GFP expression pattern (p=0.69, Fisher's exact test). (**I**) Expression of PHA-4::GFP in inverted black and white rendition at the ~100-cell stage in an equalized alive (top) and equalized dead (bottom) *lin-5(ev571)* embryo. (**J**) Quantification of embryos with abnormal PHA-4::GFP expression at the ~100-cell stage in indicated conditions. Lethality of equalized embryos is not significantly associated with PHA-4::GFP expression pattern (p=0.128, Fisher's exact test). See *Figure 5—figure supplement 1E* for detailed annotation of expression patterns.

The online version of this article includes the following source data, source code and figure supplement(s) for figure 5:

**Source code 1.** Analysis of Ea/Ep migration, cell cycle duration, and END-3::GFP expresion in lin-5(ev571) and control embryos.
**Source data 1.** Ea/Ep migration, cell cycle duration, and END-3::GFP expresion in individual *lin-5(ev571)* and control embryos.
**Figure supplement 1.** Fate acquisition in endodermal and pharyngeal lineages.

far in equalized embryos in the entire lineaging data set (Fisher's exact test), including between additional $P_4$ division, EMS skew, MSa/MSp inversion, aberrant ELT-2, or PHA-4 expression, suggesting that these defects frequently arise independently of each other.

Despite such widespread phenotypic variability among equalized *lin-5(ev571)* embryos, we wondered whether the ultimate fate of equalized embryos could be predicted early in development. Again excluding embryos with a T-arrangement at the 4-cell stage or with an EMS skew at the 6-cell stage, we utilized a machine learning approach to identify a small number of features that hold predictive value for distinguishing the two sets of embryos. To this end, we utilized Lasso (Least absolute shrinkage and selection operator), which is a penalized regression method that iteratively eliminates features that contribute the least to the predictive performance of a statistical model, eventually converging on a simpler solution (*Tibshirani, 1996*). We conducted this analysis at the 4-, 8-, 15-, and 28- cell stage, using as input all available features in each case for equalized *lin-5(ev571)* embryos (n = 31 embryos scored for all features at all four stages, *Supplementary file 7*). To obtain a robust classifier despite the relatively small sample size, we used repeated fivefold cross-validation, which randomly splits the data into five subsets of similar sizes, four of which are used for training the model, which is then evaluated against the remaining subset for goodness of fit. Performing 250 repetitions of cross-validation in each case, we found that features extracted at the 15-cell and 28-cell stage hold similarly high predictive value, with an overall model accuracy of ~88 ± 6% SD and ~85 ± 10% SD, respectively (*Figure 6A*). Remarkably, we found in particular that ~89% of the models selected at the 15-cell stage contained just three predictive features (*Figure 6B, C*): ABara dorsoventral (D-V) position, ABara positional deviation (pOV), and Ca net movement (Ca.netdis). The best performing model at the 15-cell stage achieved 100% sensitivity – meaning that all dead embryos were correctly classified – and 86% specificity – meaning that a small fraction of alive embryos were misclassified (*Figure 6D*).

## Discussion

Asymmetric division can take several forms. In some cases, distinct fates result solely from the partitioning of cytoplasmic components asymmetrically to daughter cells of equal size. In other cases, however, asymmetric division also entails different sizes of daughter cells. Here, we uncover that *C. elegans* embryos stemming from experimentally induced equalized first division exhibit a range of incompletely penetrant phenotypic deviations from the norm (*Figure 6—figure supplement 1*), demonstrating that physically asymmetric division of the zygote is

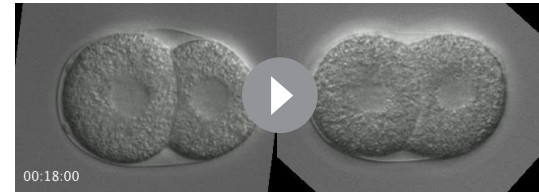

**Video 8.** Expression of the endodermal marker END-3::GFP in control (left) and equalized (right) *lin-5(ev571)* embryos (relative AB size 58% and 53%, respectively). In the equalized embryo, Ea/Ep express less END-3:: GFP than they normally do, and divide prematurely, before completing ingression.
https://elifesciences.org/articles/61714#video8

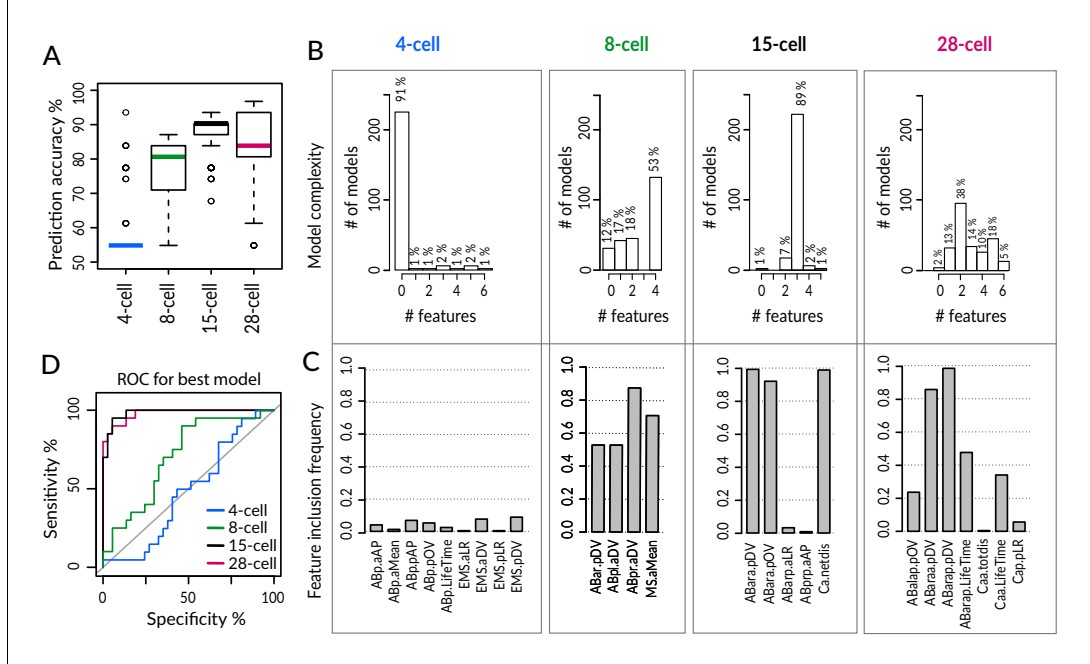

**Figure 6.** Logistic Lasso regression machine learning analysis predicts lethality in early equalized embryos. (A) Prediction accuracy of lethality versus survival in equalized embryos for individual logistic Lasso models generated by fivefold cross-validation (cv) over 250 repetitions in indicated stages. Note that embryos with a T-arrangement at the 4-cell stage or with a skewed EMS division at the 6-cell stage were excluded from this analysis, as were embryos with missing values for some of the cells. (B) Number of predictive features in individual Lasso models over cv 250 iteration; 14 alive and 17 dead equalized *lin-5*(*ev571*) embryos were used for training. Y-axis: number of iterations in which a model with the indicated number of features was selected. Zero features dominant at the 4-cell stage indicates that an empty model (intercept only) was selected in the majority of iterations and performed poorly overall. (C) Inclusion frequency of individual features in an optimal model at each cv iteration over 250 repetitions at indicated stages. Selected variables were chosen from 18, 72, 150, and 249 features present at 4-, 8-, 15-, and 28-cell stage, respectively. *Supplementary file 7* lists all features used at each stage for Lasso analysis. (D) Receiver–operator curve (ROC) of best model selected at each stage with the minimal cross-validation error. Shown are selectivity and sensitivity as a function of changing the threshold value for assigning embryos to dead or alive category; diagonal line: random classification. We noted also that even though the models at the 8-cell stage had inferior accuracy, MS division orientation was repeatedly selected among four predictive features at that stage, highlighting the importance of the MSa and MSp inversion phenotype described in *Figure 4*.

The online version of this article includes the following source data and figure supplement(s) for figure 6:

**Figure supplement 1.** Summary of phenotypes observed in equalized and inverted *lin-5*(*ev571*) embryos.

**Figure supplement 1—source data 1.** Counts of embryos with phenotypes indicated in the *Figure 6—figure supplement 1*.

**Figure supplement 2.** Comparison of cell positions at the 8- and 16-cell stage in equalized embryos.

**Figure supplement 3.** Comparison of variance and mean changes between equalized and control *lin-5*(*ev571*) embryos.

**Figure supplement 3—source data 1.** Results of statistical tests comparing means and variance between equalized and control embryos for each variable *lin-5*.

critical to ensure invariably successful embryogenesis.

## On the importance of physically unequal asymmetric cell division

A striking example of physically unequal asymmetric division is that of the female meiotic divisions, where minute polar bodies are generated in addition to the much larger oocyte. There is little doubt that the unequal size of daughter cells is of paramount importance in this case to ensure proper allocation of maternal components to the oocyte and future embryo. Physically unequal divisions, albeit less striking in terms of size inequality, are also frequent during early embryogenesis of metazoan organisms. One possibility is that this ensures swift and proper allocation of components to daughter cells destined to yield different numbers and types of descendants in early embryos that rely on maternally contributed components and undergo rapid cell cycles. This may be particularly important in holoblastic embryos, which cleave within the constraints of an eggshell or an analogous

structure, precluding compensation for potential variability in blastomere volumes through differential cell growth.

A paradigmatic example of a physically unequal division in early embryos is that of the *C. elegans* zygote, which normally yields a larger AB daughter and a smaller $P_1$ daughter. Previously, the specific role of size inequality had not been addressed independently from fate asymmetry. Here, we addressed this question by altering the function of the spindle positioning protein LIN-5 to equalize first cleavage without affecting A-P polarity. We discovered that first division equalization is tolerated to some extent, as evidenced by approximately half of equalized embryos completing development. Moreover, we discovered that embryos with a relative AB size smaller than 48% always die, revealing a size asymmetry threshold for viability.

## Alterations in cell cycle timing can be tolerated during *C. elegans* embryogenesis

Prior work established that the size difference between AB and $P_1$ contributes to their division asynchrony in the wild type because the DNA replication checkpoint is engaged preferentially in $P_1$ compared to AB (*Brauchle et al., 2003*; *Stevens et al., 2016*). In addition to this mechanism, the positive regulators of mitotic entry polo-like-kinase I (PLK-1), cyclin B3, and CDC-25 phosphatase are enriched in AB compared to $P_1$, thereby also contributing to division asynchrony, in a size-independent manner (*Budirahardja and Gönczy, 2008*; *Michael, 2016*; *Rivers et al., 2008*). As anticipated from these studies, we found here that division asynchrony decreases following first division equalization, in a manner that depends on AB size. However, this asynchrony alone does not cause subsequent death since equalized embryos with experimentally restored asynchrony die at a similar rate. This is in line with the viability of embryos depleted of DNA replication checkpoint components, which also exhibit decreased asynchrony between AB and $P_1$ (*Brauchle et al., 2003*). Likewise, the majority of embryos in which asynchrony between the two cells is abrogated entirely through local infrared treatment of $P_1$ also live (*Choi et al., 2020*). Furthermore, we uncovered here that cell cycle progression is not set merely by size in all *C. elegans* embryonic blastomeres since diminishing the size of AB and its descendants does not slow their cell cycle, perhaps because the abovementioned positive mitotic regulators are present in excess in AB.

We found also that not only $P_1$, but also many of its descendants, progress faster through the cell cycle upon first division equalization. It will be interesting to address whether these later differences reflect a sustained excess of positive regulators such as PLK-1, cyclin B3, and CDC-25 inherited following first division equalization. Regardless of the underlying cause, faster cell cycle progression in the $P_1$ lineage might have repercussions also for fate acquisition, which can be reliant on precisely timed expression of transcription factors during *C. elegans* embryogenesis (*Murray et al., 2012*; *Nair et al., 2013*; *Sarov et al., 2012*).

## Cell positioning, division orientation, and cell fate defects upon first division equalization

Although not detrimental to viability per se, faster cell cycle progression in the $P_1$ lineage, but not in the AB lineage, results over time in an altered sequence of cell division in equalized embryos, which can lead to missing or aberrant interactions between cells. A particularly striking example occurs in a minor fraction of equalized embryos that exhibit an aberrant T-arrangement at the 4-cell stage, leading to a lack of inductive Delta/Notch signaling. Suggestively, we found that equalized embryos with such a T-arrangement tend to be elongated (see legend of *Figure 2—figure supplement 1*). Compatible with embryo length playing a role, T-arrangements also occur in *lon-1* mutant embryos, which are very elongated but where AB and $P_1$ have normal relative sizes (*Yamamoto and Kimura, 2017*). Interestingly, a similar T-arrangement occurs naturally in other Nematode clades, reflecting plasticity in early division patterns that likely requires adaptations of fate determination networks (*Brauchle et al., 2009*; *Schulze and Schierenberg, 2011*; *Schulze and Schierenberg, 2009*).

Missing or aberrant interactions between cells can also derive from defective division orientation, leading to daughter cell mispositioning, as exemplified with the skewed EMS and MS division orientation. Interestingly, an EMS skew is observed also upon impaired Wnt signaling from $P_2$ (*Bei et al., 2002*; *Schlesinger et al., 1999*; *Thorpe et al., 1997*). More generally, a posterior signaling center that relies on a Wnt-dependent relay mechanism originating from $P_2/P_3$ polarizes multiple cells

during *C. elegans* embryogenesis (**Bischoff and Schnabel, 2006**), and our findings are compatible with the possibility that this signaling center is compromised upon first cleavage equalization. Furthermore, the EMS skew might affect cell–cell contacts and therefore alter the instructive Delta/Notch signaling that normally occurs between MS and two adjacent ABa-derived cells, which are thereby triggered to differentiate towards pharyngeal fates (reviewed in **Mango, 2009**). We indeed observed inappropriate induction of pharyngeal fate in several ABa cells, raising the possibility that aberrant contacts occurred between MS/MSa/MSp, which express a Delta ligand, and ABala or ABarp, which express a Notch receptor, but that are normally shielded from ligand binding owing to their distant location. Moreover, we noted that the intestinal progenitors Ea/Ep expressed less END-3::GFP in equalized embryos, perhaps reflecting insufficient induction by maternal factors, followed by stochastic expression of the downstream component ELT-2. Compatible with this view, maternally provided SKN-1, as well as its targets END-1 and END-3, must reach a critical threshold for complete activation of ELT-2 (**Maduro, 2015**; **Raj et al., 2010**).

Regarding MS position, a physical computational model predicted that abolishing both size asymmetry and asynchrony between AB and $P_1$ lineages would impact the positions of MS and E daughters at the 24- cell stage in ~25% of embryos (**Fickentscher et al., 2016**). Accordingly, we observed instances of MSa/MSp mispositioning and incomplete ingression of E daughters in equalized *lin-5 (ev571)* embryos. Thus, these defects in equalized embryos could be at least in part explained by abnormal forces experienced by cells with altered volumes and division timing.

Detailed lineage analysis enabled us to uncover that in many equalized embryos the germline progenitor $P_4$ undergoes continued divisions in a manner that resembles the behavior of its cousin D. Such a defect in germline fate specification could help explain the reduced fertility observed in some adults that derive from embryos that survived equalization. Lineage transformation of $P_4$ into D also occurs in embryos derived from *mes-1* mutant animals, in which $P_2$ and $P_3$ lack proper polarity (**Berkowitz and Strome, 2000**). By extension, such polarity defects may also be present following first division equalization. In the wild type, $P_4$ divides symmetrically and its daughters remain quiescent until hatching (**Sulston et al., 1983**). A size-dependent switch from asymmetric to symmetric division occurs in $P_4$ as it reaches a size too small to allow formation of the reciprocal PAR protein gradient (**Hubatsch et al., 2019**). Our findings lead us to speculate that $P_4$ cell size reduction might be required not only for this switch of division mode but also to maintain germline progenitors quiescent until hatching.

## Physically asymmetric division of the *C. elegans* zygote ensures invariably successful embryonic development

Wild-type *C. elegans* embryogenesis is highly stereotyped, exhibiting little variability between embryos in terms of division timing, cell positioning, division orientation, and fate acquisition, ultimately yielding 558 cells at hatching in the hermaphrodite (**Richards et al., 2013**; **Sulston et al., 1983**). The array of incompletely penetrant and variable defects revealed here implies that a physically unequal first division is critical for the stereotypy of *C. elegans* embryogenesis.

Despite important phenotypic variability upon first division equalization, we found that three features at the 15-cell stage together hold strong predictive power in predicting the ultimate fate of equalized embryos: ABara D-V position, ABara positional deviation, and Ca net displacement. Closer analysis of cell positions in the lineaging data set lends further support to the importance of these three features. Indeed, several AB descendants on the right side of embryo, including ABar and its progeny, were significantly shifted dorsally as early as the 8-cell stage solely in dying embryos (**Figure 6—figure supplement 2**, **Supplementary file 2**). Although the mechanisms responsible for such dorsally shifted position or for increased Ca movement in equalized dead *lin-5(ev571)* embryos remain to be fully understood, we conclude that these three features provide a shared early signature among most equalized embryos that then diverges into more variable phenotypic manifestations.

We also wondered whether having differently sized AB and $P_1$ blastomeres contributes to robust embryogenesis. Developmental systems are deemed to be robust if they remain unchanged or else change in a reproducible manner in the face of a given perturbation. The loss of robustness is manifested by increased variability upon perturbation without a change in the mean (reviewed in **Félix and Barkoulas, 2015**). Applying these criteria to features monitored in our data set, we found only 23 of the 1608 analyzed in which the mean remained the same but the variability increased

when comparing equalized *lin-5(ev571)* and control embryos (*Figure 6—figure supplement 3*). This indicates that these features in particular are less robust to perturbation by first division equalization. Moreover, our findings suggest that having differently sized AB and $P_1$ blastomeres provides resilience against mechanical stress, as evidenced by the increased lethality of equalized *lin-5(ev571)* embryos upon compression. Normally, compressed *C. elegans* embryos can correct specific defects in cell positioning through the concerted migration of several AB descendants (*Jelier et al., 2016*). Presumably compressed equalized embryos cannot undertake such corrective movements for steric reasons or because their fate is compromised.

How general is the importance of having differently sized blastomeres in early embryogenesis? Physically asymmetric cleavage of the zygote is present across the Rhabditida order (*Brauchle et al., 2009*; *Valfort et al., 2018*), suggesting functional importance over substantial evolutionary times. Moreover, unequal cleavages are present in early development of other systems, including annelid, ascidian, and echinoderm embryos (reviewed in *Moorhouse and Burgess, 2014*; *Negishi and Nishida, 2017*; *Shimizu et al., 1998*). Conversely, generating cells of equal sizes seems of importance during early human development since unevenly sized blastomeres frequently correlate with aneuploidy and lower rates of implantation and pregnancy (*Hardarson, 2001*). Overall, generating appropriately proportioned daughter cells is widespread in metazoan organisms, and our work demonstrates the importance of physical asymmetry of the first division in *C. elegans*, which is fundamental for ensuring invariably successful development.

## Materials and methods

### *C. elegans* strains and embryo preparation

*C. elegans* strains used in this study are listed in *Supplementary file 5* and were maintained on standard NGM plates with OP50 *Escherichia coli* as a food source. Temperature-sensitive strains were maintained at 16°C, other strains at 20°C. Strains expressing the desired fluorescent markers were crossed with the *lin-5(ev571)* carrying strain, and homozygous progeny selected based on temperature sensitivity and fluorescent marker expression. After establishing a given strain, worms were maintained for at least one generation prior to analysis. For embryo imaging, 1–2-day-old adult *lin-5 (ev571)* hermaphrodites were dissected in M9 buffer chilled at 15°C, and one-cell stage embryos with visible pronuclei collected using a mouth pipet. Embryos were then mounted either on a 2% agarose pad for DIC imaging with oil immersion objectives or into a bead slurry containing ~20 μm polystyrene beads (Polysciences, #18329-5) in M9 + 0.5% (w/v) methylcellulose for water immersion objectives or 20% iodixanol (Optiprep, Sigma Aldrich) for glycerol objectives to raise the refractive index of the medium (*Boothe et al., 2017*). Two methods were utilized to image embryos without compression. First, we mounted embryos in a viscous M9 medium containing + 0.5% (w/v) methylcellulose and 45 μm polystyrene beads. Second, embryos were mounted on a 2% agarose pad as usual, but the coverslip removed after the temperature shift, and the embryo then deposited with the pad onto an NGM plate. Non-compressed embryos obtained by either method were analyzed jointly for the extent of embryonic lethality.

### Temperature shift for size equalization

Rapid temperature shifts between 17°C and 27°C were performed on the microscope stage with the CherryTemp fluidic temperature controller (Cherry Biotech, France). The CherryTemp device is equipped with two thermalization chambers that were set to 17°C and 27°C. By changing the chamber through which the thermalization solution flowed, the sample was rapidly (~15 s) shifted to the desired temperature. To generate equally dividing *lin-5(ev571)* one-cell stage embryos, we performed the upshift right after nuclear envelope breakdown (NEBD) for embryos imaged by DIC and at metaphase for embryos expressing fluorescently labeled histone, and kept them at the restrictive temperature of 27°C until the completion of cytokinesis, ~5 min thereafter. Thereafter, the sample was shifted back to the permissive temperature of 17°C for the rest of embryogenesis. Embryos imaged only for a fraction of embryogenesis were moved to an incubator set to 17°C and survival scored the next day. In all experiments, we scored as 'alive' embryos that reached a normal looking, motile, 3-fold pre-larval stage, or that hatched, and as 'dead' embryos with an abnormal morphology at that stage or unable to hatch. Note that the *lin-5(ev571)* strain used for the majority of

experiments expressed in addition the plasma membrane marker GFP::PH and the chromatin marker mCherry::H2B; this strain exhibited a lethality on plates of 4.7% (n = 829), close to that of *lin-5 (ev571)* embryos (2.5%, n = 394).

## Time-lapse microscopy

DIC time-lapse microscopy was performed on a Zeiss Axioscope 2 equipped with DIC optics and a 100 × 1.25 NA Achrostigmat objective, recording with a USB 3.0 1.3 MP monochrome CMOS camera (Ximea, model MQ013MG-E2, Slovakia), controlled by the open-source μManager software (*Edelstein et al., 2014*).

Combined DIC and epifluorescence time-lapse microscopy was performed on a motorized Zeiss Axio Observer D1 using a 63 × 1.2 NA C-Apochromat water immersion objective, equipped with an Andor Zyla 4.2 sCMOS camera, a piezo-controlled Z-stage (Ludl Electronic Products), and an LED light source (Lumencor SOLA II). The setup was controlled by μManager.

The lineaging data set was acquired on a Leica SP8 laser scanning confocal microscope equipped with a 60× HC PL APO 1.3 NA glycerol immersion objective, HyD detectors set to 100% sensitivity, and a tunable Chameleon laser (Coherent). Time-lapse recordings were acquired at 2.5 min intervals, capturing 35 slices 0.75 μ apart, typically for 100 frames. The microscope was set to 8 kHz resonance scanning mode to reduce phototoxicity (*Richards et al., 2013*), 4 × line averaging plus 2 × frame accumulation, with a pixel size of 140 nm, a pixel dwell time of 50–70 ns, and the pinhole set to 1.2 Airy units. To compensate for the signal loss deeper in the sample, the excitation light was ramped across the Z range from 2% to 15% for the 488 nm laser line and from 2% to 18% for the 585 nm laser line. Embryos were mounted in M9–20% iodixanol (Optiprep, Sigma Aldrich) to match the sample's refractive index (*Boothe et al., 2017*) and placed in a sandwich between two #1.5 coverslips (40 × 22 mm and 18 × 18 mm) separated by ~20 μm polystyrene beads and sealed with melted VALAP (vaseline, lanolin, paraffin, 1:1:1). Sample temperature was maintained at 17°C using the CherryTemp temperature controller. The coverslip sample sandwich was attached to the CherryTemp thermalization chip glass surface through a thin layer of water.

## Optogenetic-mediated division equalization

Worms expressing LIN-5::EPDZ::mCherry, PH::EGFP::LOV, and GFP::TBB-2 in the embryo (a kind gift from Sander van den Heuvel) were dissected in a dark room under a red light to prevent premature activation of PH::LOV and mounted for imaging on 2% agarose pads (*Fielmich et al., 2018*). Imaging was performed on a Leica SP8 in the Live Data Mode of the LAS X software, allowing to toggle laser lines on/off during acquisition. The microscope settings were the same as above with the following modifications: no Z-compensation of exposure, time-lapse acquired at 10 s interval, capturing 11 slices with 1.25 μm spacing, and this for 80–100 frames (from 1- to 4-cell stage); moreover, the temperature was maintained at 22°C. LIN-5::EPDZ::mCherry distribution was monitored with a 585 nm laser; 585 nm transmitted light images were collected on an additional PMT detector to follow NEBD for analysis of AB/$P_1$ mitotic asynchrony. Interaction of LIN-5::LOV with PH::EPDZ was induced by activating the 488 nm laser at 3–5% intensity only during acquisition in a small rectangular region (~10 × 5 μm) at the anterior cortex, from NEBD until cytokinesis completion.

## Cell cycle retardation with 405 nm laser

A pulsed 405 nm diode laser (at 70% output, 700 mW/cm$^2$ at 100%) was continuously scanned over the entire $P_1$ nucleus early in the 2-cell stage for 4.5 min at 8 kHz using a 70 ns dwell time on the SP8 confocal setup described above. This induces photodamage, likely in the form of thymidine dimers in the DNA, which might cause replication fork stalling and activation of the DNA replication checkpoint. Using otherwise wild-type embryos expressing mCherry::H2B, we determined experimentally an optimal non-lethal duration (4.5 min) of continuous laser scanning that induces and enhances the delay between AB and $P_1$ mitoses from the normal ~3.5 ± 0.34 min to ~6 ± 1.6 min at 17°C.

## Image processing and analysis

All images were rotated, Z-projected, and adjusted in Fiji (ImageJ) for display (*Schindelin et al., 2012*). The 2D surface of the AB and $P_1$ cells was determined manually from DIC images or from the

GFP::PH plasma membrane signal in strains expressing this marker. This was achieved by tracing cell outlines with a Fiji polygon tool at the mid-plane, with both nuclei in focus, when the interface between AB and $P_1$ was perfectly straight, ~5 min after cytokinesis. The size of AB was then expressed relative to the embryo cross-sectional area, corresponding to the sum of the $P_1$ and AB surfaces. To assess how well the 2D mid-plane cell area measurements matched the corresponding 3D cell volumes, we analyzed embryos expressing GFP::PH, segmenting the full 3D volumes of AB and $P_1$ using a watershed-based segmentation (MorphoLibJ plugin in Fiji).

## Analysis of marker gene expression

Embryos expressing endogenously tagged PAR-2::GFP and mCherry::MEX-5 were imaged on a wide-field Zeiss Axio-Observer microscope as described above. Images were taken every 30 s until the 4-cell stage, capturing 11 planes with 1.5 µm spacing. For signal quantification, we determined the mean intensity in the mid-plane at the 2-cell stage, including the area covering the nucleus and the entire cortex. MEX-5 intensity was normalized to the mean AB intensity in control embryos from the corresponding day because MEX-5 intensity differed significantly between experiments.

Transgenic embryos expressing END-3::GFP or ELT-2::GFP in a *lin-5(ev571)* background were imaged using combined DIC and fluorescence time-lapse microscopy as described above, with 2 min intervals and 1 µm optical slicing, capturing a 25 µm stack. Gastrulation movements of Ea/Ep cells were tracked in 3D with the TrackMate Fiji plugin (*Tinevez et al., 2017*), utilizing the nuclear END-3::GFP signal, which is expressed from the beginning of the Ea/Ep cell cycle (*Maduro et al., 2005*). We then quantified the peak GFP intensity, that is, the mean voxel intensity in the nuclear volume obtained from the sphere detection macro in TrackMate. The ELT-2::GFP expression pattern was scored visually for the number of GFP-positive cells at the E8 stage. Analysis of pharyngeal differentiation was based on the PHA-4::GFP expression pattern in embryos co-expressing the pan-nuclear mCherry::H2B marker (*Sarov et al., 2012*). Embryos were imaged with the SP8 confocal microscope as described above, followed by cell lineaging as described below, allowing us to determine the identity of GFP-positive cells at the ~100-cell stage.

## Statistics and data analysis

Box and whisker plots shown throughout this article were generated in R using the boxplot function. Briefly, each box contains 50% of all data points between the first and third quartile (interquartile range [IQR]) and its center lies at the mean along the Y-axis, with the median indicated by a thick line in the box. Whiskers extend to ±1.58 IQR/sqrt(n). Data points out of this range are shown as individual points and can be considered as outliers, except for plots of variability, where outliers are not displayed for simplicity.

Statistical comparisons were performed in R using Welch's two-sample t-test with Benjamini–Hochberg correction for multiple comparisons unless indicated otherwise in the figure legends. Simultaneous comparisons of multiple groups for difference in their means were performed using Tukey's honest significant difference test.

## Lineage tracing

Three-dimensional time-lapse recordings of embryos expressing mCherry::H2B in either wild-type or *lin-5(ev571)* background were first preprocessed to enhance the nuclear signal and remove noise with the Noise2Void/CARE machine learning pipeline (*Krull et al., 2019*). Thereafter, a custom Fiji macro was used to correct the drift using the first polar body as a bright fiducial marker. The lineage was then traced using a level-set image segmentation and model evolution implemented in MATLAB (MathWorks, USA) as described previously (*Dzyubachyk et al., 2010*; *Krüger et al., 2015*), and corrected in the WormDeLux Java-based lineage editor (*Jelier et al., 2016*). Results were exported in the StarryNite format (*Bao and Murray, 2011*). Cells were then automatically named in the lineage editor according to the canonical lineage (*Sulston et al., 1983*), and manually checked afterward for possible errors, especially in mutant embryos that often substantially deviated in division orientation and cell positioning from the wild-type model used for naming.

As mentioned also in the results section, all lineage files and the source code are available at https://github.com/UPGON/worm-rules-eLife. Note that we excluded embryos with the 4-cell stage T-arrangement from the lineage analysis because they are known to exhibit specific defects

stemming from the lack of ABp induction (*Priess and Thomson, 1987*), which would introduce additional variability and thus mask potential novel phenotypes. Note that some embryos were imaged only up to the 26 cell stage; therefore, the number of embryos used for statistical comparison differs for individual cells and in different analyses reported in this article; *Supplementary file 2* reports the number of embryos/cells used for each comparison.

## Data analysis of lineaged embryos

Lineages in the StarryNite format were imported into R version 3.6.1 (*R Development Core Team, 2014*). The growth curves of all embryos were aligned in time, and a specific scaling factor for each embryo was deployed to match the mean pace of wild-type development at 17°C based on the maximum correlation between the wild type and each experimental curve (*Supplementary file 1*). Differential pace of development could be caused by slight variations in temperature or by inherent embryo variability. We then set ABa division time as time 0 because time-lapse recording for the lineaging experiments typically started at the 4-cell stage, after the temperature upshift was performed during the first division, or in the early 2-cell stage for controls.

Embryos were aligned in space with respect to their inferred A-P and D-V axes using a custom Java-based script (*Jelier et al., 2016*). First, nuclear positions were normalized to length, width, and height of each embryo along each imaging axis. Then, the alignment method first determines the A-P axis coinciding with the first principal component of the PCA calculated from all nuclear positions within first 100 frames of development. The D-V axis is orthogonal to the A-P axis and proceeds through the ventral surface of the embryo defined as the average position of the MS lineage cells. Finally, the left–right (L-R) axis is orthogonal to both the A-P and D-V axes.

## Feature extraction

We then computed 13 features (variables) using this aligned data set for each cell in every embryo. First, we computed the time of cell birth (variable name [v]: StartTime), and cell cycle duration (v: LifeTime), as well as the time of division (v: EndTime). Next, we recorded 3D positions of each cell along the three principal axes at the last frame before anaphase (variables: pAP, pLR, pDV), as well as the total trajectory distance (totdis) and the net displacement (netdis) of each cell as the Euclidean distance between nuclear 3D coordinates at birth and at the end of the cell cycle. Further, division angles (v: aAP, aLR, aDV) were computed with respect to the inferred embryonic axes at the first frame after anaphase onset. The last two variables, that is, angular (v: aMean) and positional deviation (v: pOV), required a comparison of a given embryo to the *lin-5(ev571)* control model that was determined by aligning embryos in time and space using the General Procrustes Analysis (*Gower, 1975*) and then calculating average positions of individual cells. Using this model, overall angular deviation (v: aMean) was computed as an angle between the reference division vector and the observed spindle orientation vector in a given embryo for each cell (*Jelier et al., 2016*). Similarly, positional deviation was calculated as the Euclidean distance between the position of a given nucleus at metaphase from that in the control model. *Supplementary file 2* contains complete statistics for all 13 features in experimental groups of embryos.

## Variable filtering and intra-group variation

We filtered variables according to effect size and statistical significance comparing equalized dead and alive *lin-5(ev571)* embryos at the 15-cell stage. We chose an arbitrary effect threshold of 15% and calculated Welch's unequal variances t-test for each variable. The p-value cut-off $\alpha = 0.008$ was determined for a false discovery rate of 10% using scrambled experimental data with shuffled group labels (100 repetitions). We then plotted effect size versus p-value as a volcano plot.

To assess variability within each group of embryos, we calculated the standard deviation and CV for each cell and every feature. In particular, the CV was used to compare cell cycle variability among different cells that have lineage and stage-specific differences in average cell cycle duration. To assess the mean and the variance (*Figure 6—figure supplement 3*), we required each feature to be scored in at least five embryos in both alive and dead equalized groups, thereby decreasing the total number of analyzed features to 1608.

## Predictive model of developmental outcome using Lasso

We searched the parameter space of variables at 4-, 8-, 15-, and 28-cell stage for features that would predict lethality in equalized embryos (n = 31 embryos scored for all features at all these stages). Data preprocessing before feeding data into the model building pipeline included removing those variables with low overall variance (empirically set at <2.5%, n = 24 variables), those that would have more than 25% of missing value (n = 12 variables), and also replacing missing values with the group mean (128/18,228 values), and scaling each variable within a 0–1 range. We used the penalized Lasso regression to generate a binary classification model (glmnet R package). We employed a machine learning strategy to obtain a robust value of the shrinkage coefficient lambda by performing fivefold cross-validation (cv) 250 times and choosing a model with minimal cross-validation mean squared error. We report average classification accuracy for best models selected over 250 cv iterations. To evaluate the stability of solutions thus obtained, we scored the number of chosen predictors and their identity over 250 iterations.

## Acknowledgements

We are grateful to Zoltan Spiró for his initial observation of acute temperature sensitivity of *lin-5* (*ev571*), as well as to Sander van den Heuvel, Joel Rothman, and Geraldine Seydoux for their generous sharing of strains. Some strains were provided by the Caenorhabditis Genetics Center (CGC), which is funded by the NIH Office of Research Infrastructure Programs (P40 OD010440). We thank Alessandro Berto, Alexandra Bezler, Nils Kalbfus, and Fabian Schneider for constructive comments on the manuscript, as well as Marco Mina for advice on statistics. For help with image acquisition and processing, we thank Olivier Burri and Nicolas Chiaruttini from the Bioimaging and Optics platform of the School of Life Sciences (EPFL, Lausanne).

## Additional information

### Funding

| Funder | Grant reference number | Author |
| --- | --- | --- |
| Swiss National Science Foundation | 31003A_155942 | Radek Jankele<br>Pierre Gönczy |
| Research Foundation Flanders | G055017N | Rob Jelier |

The funders had no role in study design, data collection and interpretation, or the decision to submit the work for publication.

### Author contributions

Radek Jankele, Conceptualization, Data curation, Software, Formal analysis, Investigation, Visualization, Methodology, Writing - original draft, Writing - review and editing; Rob Jelier, Conceptualization, Resources, Software, Formal analysis, Funding acquisition, Visualization, Methodology, Writing - review and editing; Pierre Gönczy, Conceptualization, Formal analysis, Supervision, Funding acquisition, Methodology, Writing - original draft, Project administration, Writing - review and editing

### Author ORCIDs

Radek Jankele https://orcid.org/0000-0002-1056-6556
Rob Jelier https://orcid.org/0000-0002-6395-1407
Pierre Gönczy https://orcid.org/0000-0002-6305-6883

### Decision letter and Author response

Decision letter https://doi.org/10.7554/eLife.61714.sa1
Author response https://doi.org/10.7554/eLife.61714.sa2

## Additional files

### Supplementary files

• Supplementary file 1. Annotation of all lineaged embryos reported in this study in *Figures 3* and *4*.

• Supplementary file 2. Statistical comparisons of all features (cell cycle timing, cell positions, division angles, migration) for individual cells between pairs of embryo groups (wild type, controls, equalized alive, equalized dead, inverted).

• Supplementary file 3. Analysis of correlation between relative AB size and cell cycle duration in all *lin-5(ev571)* embryos from *Supplementary file 1*.

• Supplementary file 4. Lst of features between 4- and 15-cell stage and comparison of mean values between equalized alive and equalized dead *lin-5 (ev571)* embryos for the volcano plot in *Figure 4— figure supplement 1A*.

• Supplementary file 5. List of *C. elegans* strains used in this study.

• Supplementary file 6. Statistical comparisons, average values, and number of observations for the figures in this study, with the exception of lineaged embryos (*Figures 3* and *4*), for which values are provided in other supplementary files.

• Supplementary file 7. List of features used for Lasso analysis at 4-, 8-, 15-, and 28-cell stage, including inclusion frequency and model coefficients for best predictive models.

• Transparent reporting form

### Data availability

All data generated or analysed during this study are included in the manuscript and supporting files. Source data files and code have been provided as individual files for: Figure 1—figure supplement 1–3, Figures 2, Figure 2—figure supplement 1, Figure 5, Figure 5—figure supplement 1, and Figure 6— figure supplement 1. Further, the lineaging data, as well as the source code used for their analysis, are available from GitHub: https://github.com/UPGON/worm-rules-eLife (copy archived at https://archive.softwareheritage.org/swh:1:rev:069c5e3147b7721885b5824282f342cac8a4de5b/).

The following dataset was generated:

| Author(s) | Year | Dataset title | Dataset URL | Database and Identifier |
|---|---|---|---|---|
| Jankele R, Jelier R, Gonczy P | 2020 | Dataset of traced lineages for embryos between 4- to 100-cell stage for Jankele et al. | https://doi.org/10.5061/dryad.ghx3ffbmx | Dryad Digital Repository, 10.5061/dryad.ghx3ffbmx |

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
