## [Decision Letter]

**Acceptance summary:**

Asymmetric cell divisions of size and fate are important in many developmental contexts. In this work the authors use a clever genetic manipulation to alter the size asymmetry of the first division of the zygote in the nematode, *C. elegans*. They find strong evidence that when the size symmetry is perturbed, the embryo experiences variation in subsequent cell divisions, cell fates, and survival, suggesting that there has been strong selection to maintain the size asymmetry of the first division in this species.

**Decision letter after peer review:**

Thank you for submitting your article "Unequal cell division of the *C. elegans* zygote buffers variability and ensures successful embryogenesis" for consideration by *eLife*. Your article has been reviewed by three peer reviewers, and the evaluation has been overseen by Michael Eisen as the Senior and Reviewing Editor. The following individuals involved in review of your submission have agreed to reveal their identity: Clemens Cabernard (Reviewer #1); Morris Maduro (Reviewer #3).

The reviewers have discussed the reviews with one another and the Reviewing Editor has drafted this decision to help you prepare a revised submission.

Summary:

Asymmetric cell division is an evolutionary conserved mechanism that creates cellular diversity. ACD can be manifested in biased segregation of molecular cell fate determinants, organelles or daughter cell size difference. Daughter cell size difference occurs frequently in metazoans but the functional significance remains poorly understood.

Here, Jankele et al., use *C. elegans* to investigate whether the stereotypic cell size difference between AB and P1 cells in the early embryo is important for overall embryo development. They use a temperature sensitive allele of Lin-5 and an optogenetic approach to misposition the mitotic spindle, resulting in an equalization of the AB and P1 cell in the early worm embryo. The authors discover that a size asymmetry threshold exists that determines embryo viability. Furthermore, they uncover that equalized embryos show faster cell cycle progression but also display additional defects in cell positioning, spindle orientation and cell fate acquisition.

This is an elegant study, addressing an important question in vivo. With its fixed cell lineage, availability of molecular genetic tools and advanced live cell imaging, *C. elegans* is the ideal choice to address this question. The use of the temperature sensitive lin-5 allele and the optogenetic approach are simple, yet effective.

In discussing the paper, the editor and reviewers had several broad concerns that must be addressed in a revision. We call particular attention to point (3) below, regarding claims of buffering, which are highlighted by the authors in the title and throughout the manuscript. It is our feeling that these claims were not made in a sufficiently rigorous manner. As the reviewers note, Barkoulas and Felix provide rigorous criteria for what should be called "robustness” and how it should be measured. It is essential that the authors either make a stronger case for robustness by the Barkoulas and Felix (or other appropriate) criteria, or tone down the claims about buffering/robustness.

Essential revisions:

1) Our first concern is that their conclusions hinge on LIN-5 manipulation only affecting size asymmetry. While they use appropriate controls, both the lin-5(ts) and the optogenetic lines already show an increase in embryonic lethality, albeit low, in control conditions (permissive temperature, no light) as well as mild cell positioning defects. Moreover, there is also a surprising amount of lethality (25%) in embryos with rather normal size asymmetry. Given the requirements for LIN-5 in many processes beyond spindle positioning in the zygote, including cortex behavior/blebbing, chromosome segregation, pronuclear rotations, spindle positioning in other cells, etc., it is difficult to rule out that these lines may be sensitized or that LIN-5 manipulation comprises development independently of size asymmetry, at least to some degree. To my knowledge, there is no other method available to reliably equalize divisions, so this is probably the best one can do at the moment. However, the authors should at least acknowledge this potential caveat in the absence of an independent mechanism to symmetrize divisions as it is very difficult to be sure that the only thing being affected in these experiments is size asymmetry.

2) LIN-5 caveats aside, while broadly consistent with a role for size asymmetry in development, what role asymmetry is playing is rather unclear. What emerges most clearly is that loss of size asymmetry via LIN-5 manipulation is associated with a highly pleiotropic array of defects, including changes in division timing, division orientation, blastomere positioning and patterns of fate induction, and it is seems that a variable number of potential defects accumulate, each potentially resulting in an additive increase in probability of death, particularly in perturbed conditions (e.g. excessive compression).

3) Finally, the phenotypic defects and variability in traits such as cell cycle timing and orientation are used to claim that size asymmetric divisions buffer variability. This is a rather stretched reading of the data. Notably, the authors cite Barkoulas and Felix, who in their review criticize the increasing frequency of claims of robustness genes/pathways. A key argument Barkoulas and Felix make is that compromising the function of a molecule in a system such that the system is more sensitive to further perturbations does not mean that molecule/pathway acts as buffering or robustness mechanism, but could simply be pushing the system closer to a breakdown point. The claim of buffering here rests on the observation that in dying embryos, phenotypes are more variable. However, as Barkoulas argue, one expects higher phenotypic variability as one moves the system away from wild-type behavior and starts to exceed the capacity of development control mechanisms to cope with defects. Following this line of thinking and given that most variability is coupled to pronounced changes in the mean behavior (i.e. shorter cell cycle times, directed positional shifts), we would argue that a more likely conclusion is that lin-5 manipulation simply introduces sub-lethal defects that prime the system for loss of viability when combined with other perturbations such as compression. Hence, while symmetrically dividing cells may be more sensitive to additional perturbations, we would argue that this result does not say anything about whether size asymmetry buffers the effects of other sources of variability or perturbations of development

4) The reviewers generally found the manuscript hard to follow, and that it would benefit from systematic efforts to make it accessible to a non-technical audience.

Reviewer #1:

1) Sibling cell size asymmetry occurs in several metazoan cell types. Unfortunately, the authors failed to cite the corresponding literature in the very first paragraph of the Introduction. I am sure this is not an attempt to inflate the novelty of this study. In most cases, the significance of daughter cell size asymmetry has not been clearly demonstrated in any organism. The authors need to reference not just what has been written in *C. elegans* on the topic but also include studies from human cells, *Drosophila* and other species.

2) The majority of the data is derived from upshifted lin-5 embryos. I wonder whether the optogenetic approach yielded the same results. It would be worth mentioning early in the manuscript why the authors chose to focus on the upshift experiments exclusively.

3) Figure labelling and annotations are often unclear. For instance, what is meant with “plates” in Figure 2D? The y-axis indicates the range of AB size in %.

4) Figure 5D and following: what do the colored bar plots on top and on the side of these scatter plots indicate? I could not find any specific information in the legend. Although I could make an educated guess, I would prefer the authors to tell me what the graphs refer to.

5) Figure 6: The authors mention the extraction of several features from their live cell imaging experiments. I was trying to find specific information about the nature of these features (the supplemental tables did not help) but did not succeed. It would be beneficial to mention in the manuscript and Figure 6B what these features are, and how and why they were extracted.

6) As mentioned above, the manuscript is riddled with imprecise and diffusive statements such as this:.…" suggesting that the mechanisms coping with equalization are not fully operative, implying that unequal first cleavage is required for invariably successful embryogenesis.” What is meant by “not fully operative”?

“Developmental systems are deemed to be robust if they remain unchanged or else change in a reproducible manner in the face of a given perturbation, with increased variability upon perturbation indicative of a non-robust system.” What is this supposed to mean?

Would it be possible to explain in simple terms what “lasso penalized logistic regression machine learning approach” entails?

7) Several conclusions appear not to be supported by the data. Examples:

– Figure 3A: The authors conclude that the differences in division timing appear not to be significant. Based on what statistical test did the authors reach this conclusion? I cannot find the corresponding information in the figure or text.

– The authors conclude that unless corrected, a skew in the EMS division axis results in embryonic lethality. Where is this shown? How can mispositioning of cells, e.g E and MS explain the observed lethality?

Reviewer #2:

Jankele et al. cleverly seeks to use two new tools, a rapidly reversible temperature sensitive allele of lin-5 and optogenetic manipulation of LIN-5 localization to equalize division of the *C. elegans* zygote and assess its impact on development through an impressive and comprehensive array of quantitative readouts, representing one of the best attempts to date in tackling the important and poorly understood question of the role of size asymmetry during cell division. The authors use this approach to make two key claims – first that division asymmetry per se is important for development, and second, that division asymmetry confers robustness to development by suppressing phenotypic variability. In general the manipulations and quantifications reflect a solid body of work and the data are consistent with a role for size asymmetry in development – though see comments above. At the same time why size asymmetry is important is less clear and we are not sold on the claims of phenotypic buffering.

Reviewer #3:

Many of the early embryonic cell divisions of *C. elegans* are asymmetrical in both cell size and fate. The first division of the zygote produces a much larger anterior AB cell and a smaller P1 cell. In this work, the authors use a clever genetic manipulation to interrupt function of the lin-5 gene product, required for the posterior displacement of the spindle that produces this asymmetric division, producing similar-sized AB and P1 cells. They can then restore LIN-5 function to subsequent divisions. An optogenetic method to enhance localization of LIN-5 to the anterior cortex could also produce a similar AB-P1 division in the absence of a temperature shift. The authors can then examine the consequences of the smaller AB and larger P1 on development.

As a control for potential disruption of A-P polarity, the authors find that asymmetric localization of MEX-5 and PAR-2 occurs apparently normally in lin-5 upshifted embryos. (They also interrupt LIN-5 function for a similar 5-minute window after a normal AB-P1 asymmetric division as a further control.)

In their first result, they find that degree of symmetry of the AB-P1 division correlates with inability of embryos to undergo proper enclosure and elongation, i.e. the more symmetrical, the more likely is inviability. In particular, if AB represents 48% of the embryo volume or smaller, embryos were always inviable. This establishes a threshold of AB/P1 size ratios compatible with viability.

The authors next examined cell division timing within the AB and P1 lineages following forced symmetry. Although correlated with loss of asymmetry in cell cycle timings, a forced delay of the AB cell cycle by mild laser treatment could restore cell cycle timings in symmetric AB/P1 embryos, but these embryos were not rescued for loss of viability. A small percentage of embryos were found in which alterations in timing resulted in a T-shaped configuration of cells at the 4-cell stage which prevents the P2-to-ABp induction required for ABp specification and is known to be lethal from other examples in which this induction is blocked.

Next, the authors perform lineage analysis up to the 120-cell stage and examine various patterns of development to understand the features that are affected by forced AB-P1 symmetry. They find that degree of symmetry correlates with cell cycle durations in the P1 lineage, but not the AB lineage. In general, the cell cycle becomes faster in the P1 descendants with the germline precursor P4 being the most affected as it also underwent an extra division. Relative cell positions between AB and P1 descendants become altered (as expected) because of the heterochrony between the two lineages, but there was not a statistically significant change in the duration of the cell cycle for individual cells between surviving and non-surviving equalized embryos, suggesting this alone was not the proximal cause of a failure to survive.

Quantitatively, the authors find that the coefficients of variation of cell cycle timing in the surviving embryo P1 lineages is smaller than in the equalized embryos that die. Hence, a loss of robustness in cell cycle durations is correlated with a lack of survival.

The authors examined their 4D lineage data at high resolution, documenting cell positions and division orientation. As might be expected, cell divisions deviated from controls over time and were more pronounced and variable in embryos that did not survive. An interesting result arose from noting that dead equalized embryos experienced more physical compression by the coverslip than controls by about 10%. More specifically, a skew in the division axis of EMS resulted in a mispositioning of MS and E that correlated with lethality. A similar mispositioning of MSa/p was correlated as well. The E daughters divide precociously if they are mis-specified, and consistent with this the authors find decreased or delayed expression of reporters that mark endoderm specification. Pharynx is made by both the MS and ABa descendants in normal embryos, and the authors find evidence that AB-derived pha-4 expression occurs in excess, likely due to disrupted MS-to-AB lineage cell-cell interactions that induced pharynx specification from the AB lineage.

Using a machine learning approach, the authors then identify (even from their small sample size) that three features (two from the AB lineage, and one from C) are sufficient to predict survival of an equalized embryo at the 15-cell stage. This is consistent with the notion that cell position deviations from normal are responsible for lethality.

I really enjoyed this paper. The experiments have a “classic” feel like the early days of experimental embryology in *C. elegans* where pressure from outside the eggshell was used to manipulate the positioning of the spindle as AB divided, reorienting ABa to contact P2 instead of ABp. The data are richly detailed and the images and analysis of high quality. The result that interfering with normal early cell divisions in a stereotyped system leads to stochastic failures in later development is not particularly novel, however given the way this question was addressed and analyzed is makes the manuscript worthy of *eLife* with addressing of my comments below.

1) If illegitimate contacts are predicted by the ectopic expression of pha-4::GFP, then it should be possible to look in the 4D data for whether or not these cells or their ancestors were in contact with MS lineage descendants, for example by measuring inter-nuclear distances between MS and AB lineage cells and comparing with distances in wild-type embryos.

2) The hypothesis that embryos tolerate deformation under a coverslip less well following perturbation of the AB-P1 asymmetry is interesting and bears further investigation. For example, it predicts that “equalized” embryos that were treated and then had the coverslip removed would be more likely to be viable.

3) A major finding of the paper is that cell cycle times in the P1 descendants are sensitive to the size of P1. In the Discussion the authors stopped short of suggesting a mechanism by which cell cycles are sped up. Are there candidate nuclear and/or cytoplasmic factors whose concentrations would be perturbed enough to speed up the cell cycle?

4) It would be good to mention findings in other nematodes that show that the alternative cell arrangements (in particular the T-arrangement) seen in the lin-5-manipulated *C. elegans* embryos are found naturally in other nematode species (e.g. Schulze and Schierenberg, 2011). The size differences between AB and P1 are also not as pronounced in other systems as in *C. elegans*. This would remind readers of the diversity of early embryonic cell arrangements in other nematodes.

5) The predictive data in the latter part of the paper could be related to studies of early human embryogenesis (e.g. Wong et al., 2010, PMID 20890283) in which features of mitosis timing and cell arrangements were used to correlate with surviving blastocysts.

6) Reduced levels of END-3::GFP could also result from a reduction of nuclear POP-1 levels in E. As a general feature of the *C. elegans* embryo, POP-1 asymmetry (see Huang et al., 2007, PMID 17567664) results from both cell-cell interactions (in some cases, like the P2 to EMS induction) and the PAR proteins: Anterior daughters resulting from A/P divisions have high POP-1 levels while posterior daughters have lower nuclear POP-1. Perturbation of division axes resulting from similar-size AB and P1 would be expected to alter POP-1 asymmetry as well. In the case of end-3, with a rotated EMS spindle, there may be higher nuclear POP-1 which would in turn decrease activation of end-3.

---

## [Author Response]

Essential revisions:1) Our first concern is that their conclusions hinge on LIN-5 manipulation only affecting size asymmetry. While they use appropriate controls, both the lin-5(ts) and the optogenetic lines already show an increase in embryonic lethality, albeit low, in control conditions (permissive temperature, no light) as well as mild cell positioning defects. Moreover, there is also a surprising amount of lethality (25%) in embryos with rather normal size asymmetry. Given the requirements for LIN-5 in many processes beyond spindle positioning in the zygote, including cortex behavior/blebbing, chromosome segregation, pronuclear rotations, spindle positioning in other cells, etc., it is difficult to rule out that these lines may be sensitized or that LIN-5 manipulation comprises development independently of size asymmetry, at least to some degree. To my knowledge, there is no other method available to reliably equalize divisions, so this is probably the best one can do at the moment. However, the authors should at least acknowledge this potential caveat in the absence of an independent mechanism to symmetrize divisions as it is very difficult to be sure that the only thing being affected in these experiments is size asymmetry.

We attempted to mitigate this potential issue by minimizing to the extent possible perturbation of LIN-5 function at other times and places. We acknowledge that the strain used for the optogenetic experiments exhibits ~23% embryonic lethality even without exposure to 488 nm laser light, indicating that mCh::LIN-5::ePDZ is not fully functional. By contrast, control *lin-5(ev571)* embryos upshifted early in the second cell cycle exhibit a marginal embryonic lethality of ~7%. Therefore, the adverse consequences of first division equalization reported in the manuscript cannot be ascribed simply to this mutant background or to spending 5 minutes at 27 ^o^C. As suggested by the reviewers, we now explicitly discuss these points in the revised manuscript.

Note also that we explored two alternative methods in attempting to symmetrize first division. First, we used a microneedle to deform the eggshell and thus prevent the posterior spindle from moving. However, when the needle was retracted in such experiments, the spindle quickly assumed a posterior position anew, thus precluding first division symmetrization. Moreover, substantial cytoplasm mixing took place in such embryos, which would have prevented rigorous analysis of further development. Second, we used laser tweezers to prevent movement of the posterior spindle pole, but the power needed to achieve this killed the embryo. Overall, these attempts led us to the conclusion that restricted perturbation of LIN-5 function, using primarily the *lin-5(ev571)* allele, was best suited for analyzing the consequences of first division symmetrization for subsequent development.

2) LIN-5 caveats aside, while broadly consistent with a role for size asymmetry in development, what role asymmetry is playing is rather unclear. What emerges most clearly is that loss of size asymmetry via LIN-5 manipulation is associated with a highly pleiotropic array of defects, including changes in division timing, division orientation, blastomere positioning and patterns of fate induction, and it is seems that a variable number of potential defects accumulate, each potentially resulting in an additive increase in probability of death, particularly in perturbed conditions (e.g. excessive compression).

We fully agree with this view: loss of size asymmetry indeed results in a pleiotropic array of defects rather that a single phenotype. Figure 6—figure supplement 1 encapsulates this point, which we now emphasize also in the Abstract, and expand on in the Discussion section of the revised manuscript.

In addition, we have now analyzed the phenotype of those equalized *lin-5(ev571)* embryos that live beyond embryogenesis, finding additional defects in larvae and adults. This data set is reported in the novel panel Figure 1—figure supplement 3C, and indicates that first division equalization can be detrimental also beyond embryogenesis, a point that is mentioned also in the revised manuscript.

3) Finally, the phenotypic defects and variability in traits such as cell cycle timing and orientation are used to claim that size asymmetric divisions buffer variability. This is a rather stretched reading of the data. Notably, the authors cite Barkoulas and Felix, who in their review criticize the increasing frequency of claims of robustness genes/pathways. A key argument Barkoulas and Felix make is that compromising the function of a molecule in a system such that the system is more sensitive to further perturbations does not mean that molecule/pathway acts as buffering or robustness mechanism, but could simply be pushing the system closer to a breakdown point. The claim of buffering here rests on the observation that in dying embryos, phenotypes are more variable. However, as Barkoulas argue, one expects higher phenotypic variability as one moves the system away from wild-type behavior and starts to exceed the capacity of development control mechanisms to cope with defects. Following this line of thinking and given that most variability is coupled to pronounced changes in the mean behavior (i.e. shorter cell cycle times, directed positional shifts), we would argue that a more likely conclusion is that lin-5 manipulation simply introduces sub-lethal defects that prime the system for loss of viability when combined with other perturbations such as compression. Hence, while symmetrically dividing cells may be more sensitive to additional perturbations, we would argue that this result does not say anything about whether size asymmetry buffers the effects of other sources of variability or perturbations of development

We thank the reviewers for bringing up this important point. We agree that our original conclusions attributing the observed increase in variance upon first division equalization to loss of robustness did not fit the strict definition of Barkoulas and Félix, which requires increased variance without a change in the mean upon a challenge. As a result, we altered substantially the manuscript, including changing the title to “Physically asymmetric division ensures invariably successful embryogenesis”.

Moreover, we removed statements about robustness from the Results section entirely, focusing instead solely on the observed increases in variance. We also analyzed our data further, finding that only a minority of features (23/1608 between the 4- and 100- cell stage) exhibited a significant increase of variance without a change in the mean in equalized embryos, suggestive of loss of robustness upon first division equalization merely for these features. We report these findings in the new Figure 6—figure supplement 3, mentioning them also briefly in the Discussion.

4) The reviewers generally found the manuscript hard to follow, and that it would benefit from systematic efforts to make it accessible to a non-technical audience.

We have reorganized and streamlined the manuscript so that it is now hopefully accessible to a broader readership.

Reviewer #1:1) Sibling cell size asymmetry occurs in several metazoan cell types. Unfortunately, the authors failed to cite the corresponding literature in the very first paragraph of the Introduction. I am sure this is not an attempt to inflate the novelty of this study. In most cases, the significance of daughter cell size asymmetry has not been clearly demonstrated in any organism. The authors need to reference not just what has been written in *C. elegans* on the topic but also include studies from human cells, *Drosophila* and other species.

We had cited such analyzes of cell size asymmetry in other metazoan systems in the Discussion of the original submission, but agree that they could be mentioned earlier in the text. Therefore, we moved this entire paragraph to the Introduction of the revised manuscript.

2) The majority of the data is derived from upshifted lin-5 embryos. I wonder whether the optogenetic approach yielded the same results. It would be worth mentioning early in the manuscript why the authors chose to focus on the upshift experiments exclusively.

We chose to focus on the upshift experiments with *lin-5(ev571*) for two reasons. First, as mentioned above, the optogenetic strain exhibits ~23% embryonic lethality even without exposure to 488 nm laser light, which would have hampered analyzing the consequences of first division equalization for subsequent development. Second, the optogenetic strain is not amenable to investigation with most extant fluorescently labeled marker strains because both GFP and mCherry channels are already occupied with PH::eGFP::LOV and LIN5::ePDZ::mCherry, something that was already mentioned in the Materials and methods section of the original submission. These two reasons are now mentioned explicitly in the Results section of the revised manuscript. Importantly, in addition, we now report that the terminal defective ventral closure phenotype of embryos equalized using optogenetics was similar to that observed in equalized *lin-5(ev571)* embryos (with the exception of one embryo that reached the 3-fold state but did not hatch), as mentioned in the revised manuscript.

3) Figure labelling and annotations are often unclear. For instance, what is meant with “plates” in Figure 2D? The y-axis indicates the range of AB size in %.

The legend for Figure 2D clearly states that “Plates” refers to the hatching rate on plates of embryos expressing LIN-5::ePDZ::mCherry and PH::eGFP::LOV. To further help clarify that both “Plates” and “No light” correspond to control conditions, we separated them with a horizontal line in the revised Figure 2D to prevent confusion with the y-axis label “AB size %”.

4) Figure 5D and following: what do the colored bar plots on top and on the side of these scatter plots indicate? I could not find any specific information in the legend. Although I could make an educated guess, I would prefer the authors to tell me what the graphs refer to.

Apologies about this lack of clarity. We added an explanation regarding these marginal boxplots in the legend of Figure 5. Note also that we imaged additional embryos expressing END-3::GFP, thereby increasing the data set for Figure 5.

5) Figure 6: The authors mention the extraction of several features from their live cell imaging experiments. I was trying to find specific information about the nature of these features (the supplemental tables did not help) but did not succeed. It would be beneficial to mention in the manuscript and Figure 6B what these features are, and how and why they were extracted.

We thank the reviewer for his comment. The features in question correspond to all measured variables for each cell, i.e. the time of division, cell cycle duration, cell position in 3D, division angles along the 3 embryonic axes, as well as two derived measures compared to control embryos – cell mispositioning and angular deviation of division axis. We clarified the nature of these features in the revised manuscript by expanding the corresponding Material and Methods section, and generated a new Supplementary file 7 listing all the features used at each stage, as well as inclusion frequency during model selection and coefficients for the best performing model.

6) As mentioned above, the manuscript is riddled with imprecise and diffusive statements such as this:.…" suggesting that the mechanisms coping with equalization are not fully operative, implying that unequal first cleavage is required for invariably successful embryogenesis.” What is meant by “not fully operative”?

We modified the sentence that was singled out, thereby clarifying what was meant. We have likewise altered the writing in other places to minimize and hopefully eliminate what was perceived as imprecise and diffusive statements.

“Developmental systems are deemed to be robust if they remain unchanged or else change in a reproducible manner in the face of a given perturbation, with increased variability upon perturbation indicative of a non-robust system.” What is this supposed to mean?

What we intended to convey is that developmental robustness refers to the capacity of systems to produce a uniform output despite internal noise and changing external conditions. As mentioned above, loss of robustness is characterized by increased variance in a given trait without change of mean value upon perturbation. As also mentioned above, we have revised our conclusions regarding robustness, which has led to the removal of the sentence singled out by the reviewer from the revised manuscript.

Would it be possible to explain in simple terms what “lasso penalized logistic regression machine learning approach” entails?

Following the request of the reviewer, we now explain that Lasso (Least Absolute Shrinkage and Selection Operator) is a form of penalized regression approach that iteratively eliminates features that contribute least to a predictive performance of a statistical model, thus eventually converging on a simpler solution.

7) Several conclusions appear not to be supported by the data. Examples:– Figure 3A: The authors conclude that the differences in division timing appear not to be significant. Based on what statistical test did the authors reach this conclusion? I cannot find the corresponding information in the figure or text.

We compared cell cycle duration of all individual cells between all pairwise combinations of experimental groups (i.e. wild-type, control, equalized alive, equalized dead, inverted) with Welsh’s two sample t-test with Benjamini-Hochberg correction. We added a reference in the text to Supplementary file 2, which contains all the statistical analysis for the lineaging dataset, including division timing of individual cells. This shows notably that there is no statistically significant change in timing comparing equalized dead and alive embryos. In addition, we now mention explicitly in the text that the confidence intervals of all groups shown in Figure 3A fully overlap, indicating that there is no difference between proliferation curves for AB lineage cells.

– The authors conclude that unless corrected, a skew in the EMS division axis results in embryonic lethality. Where is this shown? How can mispositioning of cells, e.g E and MS explain the observed lethality?

Our statement “unless corrected” stems from the data reported in Figure 4F, where it can be seen that four equalized embryos with an initial EMS division angle deviation >35º survived (Eq. Alive) and, as revealed upon closer inspection of the corresponding lineaging data, exhibited normal E and MS positions following correction during late anaphase/telophase. The lineage data supporting this and all other observations are available from GitHub (https://github.com/UPGON/worm-rules-*eLife*). That this is the case was already indicated in the Materials and methods section of the original submission, but is now mentioned also explicitly in the Results section of the revised manuscript.

Furthermore, we now mention a potential hypothesis regarding how such a skew, if not corrected, could eventually lead to embryonic lethality.

Reviewer #3:[…]1) If illegitimate contacts are predicted by the ectopic expression of pha-4::GFP, then it should be possible to look in the 4D data for whether or not these cells or their ancestors were in contact with MS lineage descendants, for example by measuring inter-nuclear distances between MS and AB lineage cells and comparing with distances in wild-type embryos.

We thank the reviewer for suggesting this interesting analysis. We investigated inter-nuclear distances between MS and ABal/ABar, as well as between their descendants, in equalized embryos expressing PHA-4::GFP (data from Figure 5G, code included on GitHub).

The most frequently observed aberrant pattern in our data was the ectopic expression of PHA-4::GFP in ABala lineage (see Figure 5—figure supplement 1). Therefore, we compared the 5 such embryos to the 14 that did not express GFP in any ABala cell. We found that MS-ABal, MS-ABala and MSa-ABala internuclear distances were not reduced (Wilcoxon test, not shown). However, we found that MS-ABara and MSa-ABara were significantly further apart in embryos with ectopic PHA4::GFP (see Author response image 1), for reasons that remain to be determined. Given the already dense contents of our work, we chose not to include this piece of information in the revised manuscript.

2) The hypothesis that embryos tolerate deformation under a coverslip less well following perturbation of the AB-P1 asymmetry is interesting and bears further investigation. For example, it predicts that “equalized” embryos that were treated and then had the coverslip removed would be more likely to be viable.

This comment is related to the point raised by reviewer 2, which led us to perform experiments under conditions in which embryos were not compressed. As mentioned above, those experiments lead us to conclude that compression further challenges embryos in which AB size is diminished, as evidenced by compounded embryonic lethality.

Prompted by the suggestion of the reviewer, we removed the coverslip at the 2- to 4-cell stage to recover specimens for scoring of potential post-embryonic phenotypes. In such an uncompressed setting, we observed 35% lethality among equalized embryos (N=20), which was not significantly different from the 52% lethality in compressed equalized embryos reported in Figure 2A (N=174, p=0.24, Fisher-exact test, documented in Figure 2—source code 1, and source data 1), due to the small sample size. However, combining this small data set with that of embryos imaged using 45 μm beads, we found that lethality decreased in a statistically significant manner to 33% in uncompressed equalized embryos (AB size range 48-53%, N=174 and 45, respectively, p = 0.03, Fisher’s exact test). Overall, these findings indicate that mechanical constrains imposed by compression make equalized embryos more prone to failure during embryonic development, and are reported in the revised manuscript.

3) A major finding of the paper is that cell cycle times in the P1 descendants are sensitive to the size of P1. In the Discussion the authors stopped short of suggesting a mechanism by which cell cycles are sped up. Are there candidate nuclear and/or cytoplasmic factors whose concentrations would be perturbed enough to speed up the cell cycle?

Potential candidates include PLK-1, cyclin B3 and the phosphatase CDC-25, which are positive regulator of cell cycle progression and are normally enriched in AB compared to P_1_. Following first division equalization, P_1_ and its descendants are predicted to inherit more such components than normally owing to more centrally positioned cleavage, which could promote cell cycle progression. This hypothesis has been added to the Discussion of the revised manuscript.

4) It would be good to mention findings in other nematodes that show that the alternative cell arrangements (in particular the T-arrangement) seen in the lin-5-manipulated *C. elegans* embryos are found naturally in other nematode species (e.g. Schulze and Schierenberg, 2011). The size differences between AB and P1 are also not as pronounced in other systems as in *C. elegans*. This would remind readers of the diversity of early embryonic cell arrangements in other nematodes.

Thank you for these suggestions, which we took on board. We now mention these notions in the revised manuscript.

5) The predictive data in the latter part of the paper could be related to studies of early human embryogenesis (e.g. Wong et al., 2010, PMID 20890283) in which features of mitosis timing and cell arrangements were used to correlate with surviving blastocysts.

We thank the reviewer for pointing out to this interesting paper. However, after careful reading, we found two important differences between that study and our work. First, the authors used three time-related variables for predicting blastocyst survival, but did not factor blastomere size in their model. An examination of the Lasso analysis reported in our Figure 6 shows that variables related to cell cycle timing rarely figure among good predictors. Second, the work of Wong et al. demonstrates that all the differences observed amongst human embryos can be traced to the quality of the oocyte, while our experimental intervention occurs in the zygote. Interestingly, in addition, a study by Hardarson et al., 2001 found that having unevenly sized blastomeres in early human embryos frequently correlates with aneuploidy and lower rates of implantation and pregnancy, indicating that proper size allocation at cell division is also important in that case, albeit in a different manner than in *C. elegans*. That this is the case is mentioned in the Discussion of the revised manuscript.

6) Reduced levels of END-3::GFP could also result from a reduction of nuclear POP-1 levels in E. As a general feature of the *C. elegans* embryo, POP-1 asymmetry (see Huang et al., 2007, PMID 17567664) results from both cell-cell interactions (in some cases, like the P2 to EMS induction) and the PAR proteins: Anterior daughters resulting from A/P divisions have high POP-1 levels while posterior daughters have lower nuclear POP-1. Perturbation of division axes resulting from similar-size AB and P1 would be expected to alter POP-1 asymmetry as well. In the case of end-3, with a rotated EMS spindle, there may be higher nuclear POP-1 which would in turn decrease activation of end-3.

We agree with the reviewer that it would be interesting to assess nucler POP-1 levels in equalized embryos, but this is something that will have to be left out for the future owing to time constraints.